# Masked Gated Linear Unit

**Yukito Tajima**[1]    **Nakamasa Inoue**[1]    **Yusuke Sekikawa**[2]    **Ikuro Sato**[1,2]    **Rio Yokota**[1]

[1]Institute of Science Tokyo, Japan    [2]Denso IT Laboratory, Japan

yukito@rio.scrc.iir.isct.ac.jp

## Abstract

Gated Linear Units (GLUs) have become essential components in the feed-forward networks of state-of-the-art Large Language Models (LLMs). However, they require twice as many memory reads compared to feed-forward layers without gating, due to the use of separate weight matrices for the gate and value streams. To address this bottleneck, we introduce Masked Gated Linear Units (MGLUs), a novel family of GLUs with an efficient kernel implementation. The core contribution of MGLUs include: (1) the Mixture of Element-wise Gating (MoEG) architecture that learns multiple binary masks, each determining gate or value assignments at the element level on a single shared weight matrix resulting in reduced memory transfer, and (2) FlashMGLU, a hardware-friendly kernel that yields up to a $19.7\times$ inference-time speed-up over a naïve PyTorch MGLU and is 47% more memory-efficient and 34% faster than standard GLUs despite added architectural complexity on an RTX5090 GPU. In LLM experiments, the Swish-activated variant SwiMGLU preserves its memory advantages while matching—or even surpassing—the downstream accuracy of the SwiGLU baseline.

## 1   Introduction

Transformers (Vaswani et al., 2017) have revolutionized deep learning and given rise to large-scale language models (LLMs) that achieve remarkable results across a wide spectrum of natural language processing tasks (Brown et al., 2020; Wei et al., 2022; Ouyang et al., 2022; Grattafiori et al., 2024; OpenAI, 2024; Gemma Team, 2025). Yet the explosive growth in parameter count and inference scaling translates directly into substantial inference cost and latency. The decode latency is fundamentally dominated by communication between high-bandwidth memory (HBM) and SRAM due to transferring model weights for computation, therefore memory reduction being crucial in real-world applications (Gholami et al., 2024).

Modern LLMs predominantly adopt a decoder-only architecture (Radford et al., 2018), with each decoder layer typically consisting of two modules: a self-attention module and a feed-forward network (FFN). Recent successful optimizations for attention mechanisms, such as FlashAttention (Dao et al., 2022; Dao, 2024; Shah et al., 2024) and alternative approaches like the Mamba (Gu and Dao, 2025; Dao and Gu, 2024), have significantly improved the computational efficiency by reducing glboal memory reads/writes. Nevertheless, kernel-level optimization of the FFN remains particularly challenging, as its simple architecture provides fewer opportunities for efficiency improvements compared to attention.

The activation function within the FFN plays a crucial role in determining the decoder output. Traditional nonlinearities such as ReLU (Nair and Hinton, 2010) and GELU (Hendrycks and Gimpel, 2016) laid the groundwork for deep learning, but the Gated Linear Unit (GLU) variants (Dauphin et al., 2017a; Shazeer, 2020) introduce multiplicative interactions that markedly boost expressivity. For example, SwiGLU, which swaps the sigmoid gate in the original GLU for the smoother Swish function (Ramachandran et al., 2017), has become a primary choice in modern LLMs because it

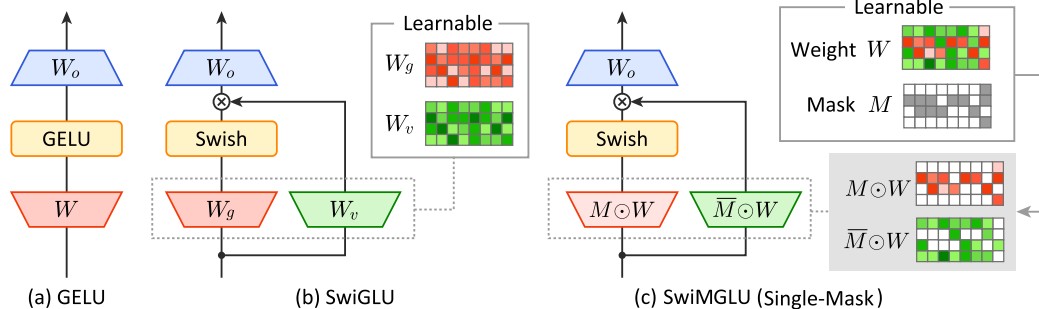

Figure 1: Comparison of FFNs. (a) Two-layer FFN using GELU. (b) SwiGLU FFN with a gating mechanism that requires two separate weights: $W_g$ and $W_v$. (c) Single-mask variant of SwiMGLU FFN (ours), which introduces a learnable binary mask $M$ to decompose a single weight $W$ into two complementary projections, reducing the required memory load during inference.

further enhances convergence and downstream accuracy. However, the explicit gating operations involving two separate projections and their and associated memory overhead at inference time can erode throughput gains, especially in latency-sensitive scenarios.

In this paper, we propose **Masked Gated Linear Units (MGLUs)**, a novel family of GLUs, along with an efficient CUDA kernel implementation. Rather than maintaining two separate projections $W_g, W_v$ for the gate and value streams, our MGLUs adopt the Mixture of Element-wise Gating (MoEG) architecture, which applies learnable binary masks $M_i$ to a single projection $W$, reproducing GLU's hallmark gating interactions. In Figure 1, we illustrate a single-mask variant, where a single weight $W$ performs both gate and value projections via a mask $M$.

The core mechanism enabling efficient kernel implementation, which we call **FlashMGLU**, lies in using complementary masks $\overline{M}_i = \mathbf{1} - M_i$. Given an input vector $\boldsymbol{x}$, FlashMGLU simultaneously computes the gate $\boldsymbol{x}(M_i \odot W)$ and the value $\boldsymbol{x}(\overline{M}_i \odot W)$ by leveraging their complementarity. With four masks and a Swish activation applied, our SwiMGLU variant achieves downstream performance comparable to or even better than the widely adopted SwiGLU, while reducing computational cost of the projection layers by 29.1% and memory usage by 37.5% during inference. It is worth noting that FlashMGLU is $12.51\times$ faster than a naïve PyTorch implementation of MGLU, opening new opportunities for more efficient and effective implementations of FFNs.

Our primary contributions are summarized as follows:

- We propose **MGLUs**, a novel family of GLUs with the MoEG architecture. We effectively mimic and enhance gating mechanism without incurring the computational and memory overhead associated with two separate full-rank projections.
- We introduce **FlashMGLU**, an efficient CUDA kernel implementation of MGLUs, reducing memory bandwidth requirements and enabling faster inference with minimal code modification.
- We conduct extensive experiments on a variety of downstream NLP tasks, demonstrating that SwiMGLU achieves comparable or superior downstream accuracy to SwiGLU while notably improving inference throughput and memory efficiency, validating its practical effectiveness for resource-constrained LLM deployments.

## 2 Related Work

**Efficeint Inference via Sparse Masks.** Model pruning methods seek to reduce the inference memory load by eliminating redundant weights. Early unstructured pruning methods removes individual parameters to achieve high sparsity (Frankle and Carbin, 2019; Xia et al., 2022). While unstructured pruning yields irregular patterns that impede efficient execution, structured pruning excises entire components for straightforward speedups at the cost of coarse-grained weight removal (Hou et al., 2025; Sandri et al., 2025; Ashkboos et al., 2024). Semi-structured approaches blend these two extremes by enforcing regular N:M sparsity blocks, combining fine-grained flexibility with hardware-friendly patterns (Fang et al., 2024; Sun et al., 2024; Frantar and Alistarh, 2023). Learnable N:M domain specific specialized masks optimized on calibration data to attain up to 50% sparsity with minimal perplexity degradation and marked improvements in inference throughput. However, the

weights that are masked out still occupy storage but take no part in computation, leaving a reservoir of unused capacity. If we can reactivate or repurpose these dormant parameters in a controlled way, we may further boost accuracy without increasing the deployed model's memory footprint.

**Efficeint Inference via Activation Sparsity.** Whereas weight sparsity reduces model size, activation sparsity directly lowers runtime FLOPs at inference (Zhang et al., 2025; HAZIZA et al., 2025; Mirzadeh et al., 2024). DejaVu shows that, for each input, only a small, input-dependent subset of heads and MLP channels is required; a lightweight predictor selects them on-the-fly, halving latency without pre-training or quality loss (Liu et al., 2023). Complementary to this contextual approach, TEAL applies magnitude-based pruning to the hidden states themselves, achieving 40–50% uniform sparsity without any retraining and realizing decoding speed-ups on modern architectures (Liu et al., 2025). Activation sparsity methods reduce computation and memory load by skipping unneeded activations during inference. In contrast we compress weight matrices directly by adding arithmetic complexity in order to generate multiple streams of outputs.

**Choices of Activation Functions in Large Language Models.** Transformer (Vaswani et al., 2017) MLP/FFN layers initially adopted simple piecewise-linear rectifiers such as ReLU (Nair and Hinton, 2010), and Gaussian Error Linear Units (GELUs) (Hendrycks and Gimpel, 2016) which weight inputs by the Gaussian CDF to produce smooth, non-saturating transitions. Later, non-monotonic self-gating activations like Swish ($x \cdot \text{sigmoid}(x)$) were discovered via automated search, offering improved deep network performance through enhanced gradient propagation (Ramachandran et al., 2017). More expressive gated linear units (GLUs) introduce multiplicative interactions by splitting a projection into value and gate streams modulated via sigmoid (Dauphin et al., 2017b; Shazeer, 2020); variants such as GEGLU (using GELU) and SwiGLU (using Swish) have been incorporated into Llama series (Grattafiori et al., 2024) to boost convergence and downstream accuracy (Shazeer, 2020).

## 3 Method

This section presents the **Masked Gated Linear Units (MGLUs)**, a novel family of GLUs that reduces memory access by emulating the gate and value streams using a single shared weight matrix. We first review the GLU variants and discuss the challenges associated with sharing their two separate projections. We then introduce MGLUs, which replace explicit gating with a mixture of element-wise gating (MoEG), sculpting distinct subspaces from the single weight matrix through learnable binary masks. Although our MoEG increases the architectural complexity, it is designed to enable efficient CUDA kernel implementation, eliminating extra matrix multiplies and memory accesses.

### 3.1 Preliminary

**GLU Variants.** The GLU variants for FFNs (Dauphin et al., 2017a; Shazeer, 2020) augment a standard two-layer FFN by splitting the intermediate projection into the gate and value streams. Specifically, given an input vector $x \in \mathbb{R}^h$ and two learnable weight matrices $W_g, W_v \in \mathbb{R}^{h \times d}$, the generalized GLU layer[1] is defined as:

$$\text{GLU}(x) \ = \ g\big(xW_g\big) \odot \big(xW_v\big), \tag{1}$$

where $g$ is a gating function, $\odot$ denotes Hadamard product, $h$ is the hidden size, and $d$ is the intermediate size. The resulting intermediate representation is then mapped back to the hidden size by an output projection $W_o \in \mathbb{R}^{d \times h}$. Figures 1(a) and 1(b) illustrate the architectures of the standard two-layer Linear Unit (LU) FFN using GELU (Hendrycks and Gimpel, 2016) and the SwiGLU FFN using Swish (Ramachandran et al., 2017) for $g$, respectively.

**Can two matrices be shared in SwiGLU?** While SwiGLU is a primary choice in state-of-the-art LLMs (Grattafiori et al., 2024), the separate value and gate projection matrices incur $2\times$ the memory reads compared to a single linear layer. One might ask whether a single shared weight $W$ can serve both streams, for example by using distinct channel-wise transformations or learned offsets. However, naively sharing $W$ typically collapses the expressivity of the multiplicative interaction: independent full-rank projections are required

| FFN | Parameters | PPL |
|---|---|---|
| GELU | $W$ | 25.6 |
| SwiGLU | $\{W_g, W_v\}$ | **23.6** |
| SwiGLU (shared) | $W_g = W_v$ | 27.0 |

Table 1: Perplexity comparison.

---

[1]Throughout this paper, we refer to the generalized GLU layer simply as the GLU layer unless ambiguity arises. Following recent practice in LLMs, we omit bias vectors, but incorporating them is straightforward.

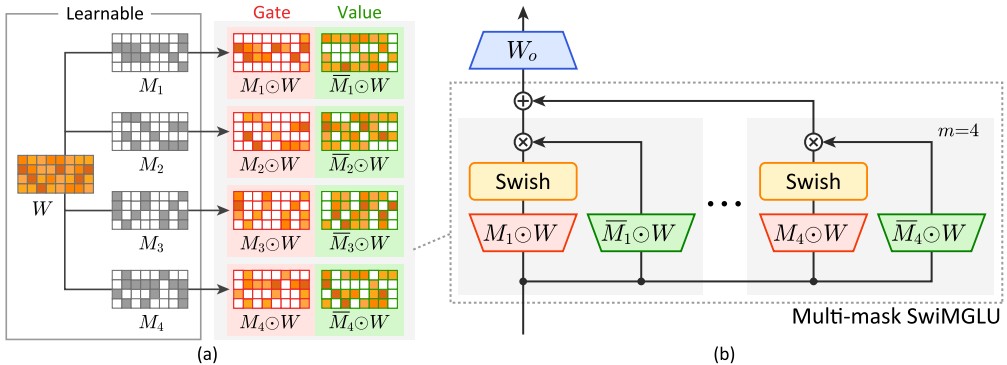

Figure 2: **Mixture of Element-wise Gating (MoEG).** (a) All gate and value projection matrices are computed from the shared weight matrix $W$. (b) The MoEG-based SwiMGLU architecture with $m$ routes, each of which leverages element-wise gating.

to learn disentangled feature gating and value transformations as shown in Table 1. This could be a limitation of explicit gating, motivating us to propose MGLUs with element-wise gating.

### 3.2 MGLU Layer

Our goal is to retain the expressive multiplicative effect of the GLU layer while reducing the number of its full-rank projections. Instead of the two separate projection matrices $\{W_g, W_v\}$, our MGLU layer introduces a small set of binary masks $\mathcal{M} = \{M_i\}_{i=1}^{n_m}$, each of which sculpts different subspaces of a single weight matrix $W$.

**Single-Mask Variant.** We start by describing the single-mask variant of the MGLU layer. Let $W \in \mathbb{R}^{h \times d}$ be a shared weight matrix and $M \in \{0, 1\}^{h \times d}$ be a binary mask matrix. We define the MGLU layer as

$$\mathrm{MGLU}_1(\boldsymbol{x}) = g\big(\boldsymbol{x}(M \odot W)\big) \odot \big(\boldsymbol{x}(\overline{M} \odot W)\big), \tag{2}$$

where $\boldsymbol{x}$ is an input vector, $g$ is a gating function, and $\overline{M} = \mathbf{1} - M$ is the complementary mask. As shown in Figure 1(c), the MGLU layer recovers the multiplicative gate-value interaction using only one full-rank weight matrix $W$. Through element-wise gating with two binary matrices $\{M, \overline{M}\}$, $W$ is effectively partitioned into two complementary subspaces, resulting in greater parameter efficiency compared to the GLU layer. During training, $M$ is optimized jointly with $W$ via the straight-through estimator (Bengio et al., 2013) on the binarization.

We utilize the complementary mask rather than two separate masks because it enables a more efficient CUDA kernel implementation. FlashMGLU in Section 4 computes the gate projection $\boldsymbol{x}(M \odot W)$ and the value projection $\boldsymbol{x}(\overline{M} \odot W)$ simultaneously by leveraging their complementarity.

**MoEG Variant.** To capture diverse gating patterns across channels, the MoEG architecture allows multiple complementary masks and lets the model learn which subspace of $W$ serves as gate versus value. Specifically, we define the MoEG-based MGLU as follows:

$$\mathrm{MGLU}_{n_m}(\boldsymbol{x}) = \sum_{i=1}^{n_m} \Big[ g\big(\boldsymbol{x}(M_i \odot W)\big) \odot \big(\boldsymbol{x}(\overline{M_i} \odot W)\big) \Big], \tag{3}$$

where $n_m$ is the number of mixtures, and $\mathcal{M} = \{M_i\}_{i=1}^{n_m}$ is a set of binary mask matrices.

Figure 2 shows the overall architecture. As shown, all gate projection matrices $M_i \odot W$ and value projection matrices $\overline{M_i} \odot W$ are derived from the shared full-rank weight matrix $W$ (Figure 2(a)), and their resulting representations are subsequently aggregated (Figure 2(b)). This architecture, featuring $m$ parallel routes, can be interpreted as a variant of the mixture of expert architecture (Jacobs et al., 1991; Jordan and Jacobs, 1994), thereby improving representational capacity. During training, the masks $\{M_i\}_{i=1}^{n_m}$ are optimized jointly with $W$. At inference, all masks are fixed, and the fused masked projections execute with a single kernel.

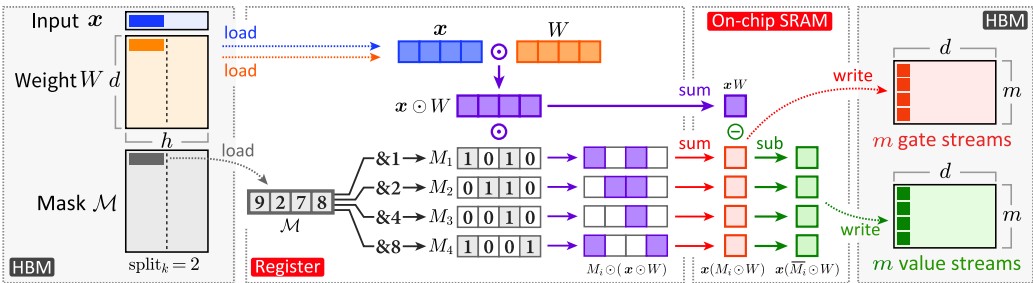

Figure 3: Diagram of how **FlashMGLU** forward pass is performed during generative inference. When the weight $W$ is split into two blocks in the $K$ dimension ($\text{split}_k = 2$), the input vector $x$ is also partitioned into two. We pre-compute the unmasked matrix-vector operation, than selectively add-up the sum according to the mask values avoiding excessive memory reads of weight matrices.

---

**Algorithm 1 FlashMGLU forward pass:** Split-K Matrix Vector Product with Packed $n_m$ Masks.

---

**Require:** $A \in \mathbb{R}^{M \times N}$, $x \in \mathbb{R}^N$, $\text{mask} \in \{0, \ldots, 2^{n_m} - 1\}^{M \times N}$ on HBM, $\text{split}_k$
1: Initialize accumulators $z \in \mathbb{R}^{2\,n_m \times M} \leftarrow 0$
2: **for** $\text{row} = 0$ **to** $M - 1$ **do**
3:     **for** $\text{chunk} = 0$ **to** $\text{split}_k - 1$ **do**
4:         $s \leftarrow \lceil (N + \text{split}_k - 1)/\text{split}_k \rceil$, $\text{start} \leftarrow \text{chunk} \times s$, $\text{end} \leftarrow \min(\text{start} + s, N)$
5:         Each thread sets $t \leftarrow 0$ and $s_i \leftarrow 0$ for $i = 1, \ldots, n_m$
6:         **for** each $k = \text{start}, \ldots, \text{end} - 1$ (stride $= \text{blockDim}.x$) **do**
7:             Load $A[\text{row}, k], x[k], \text{mask}[\text{row}, k]$ from HBM to register.
8:             On chip, compute $v = A[\text{row}, k] \times x[k]$
9:             On chip, compute $t = t + v$
10:             **for** $i = 1, \ldots, n_m$ **do**
11:                 On chip, compute if $\text{mask}[\text{row}, k] \wedge (1 \ll (i - 1)) \neq 0$ then $s_i = s_i + v$
12:             **end for**
13:         **end for**
14:         Reduce $t$ and all $s_i$ across threads on SRAM.
15:         **for** $i = 1, \ldots, n_m$ **do**
16:             $\text{atomicAdd}\big(z[i, \text{row}],\ s_i\big), \text{atomicAdd}\big(z[n_m + i, \text{row}],\ t - s_i\big)$ to HBM.
17:         **end for**
18:     **end for**
19: **end for**
20: **return** $z$

---

# 4 FlashMGLU: Efficient Kernel Implementation of MGLUs

Hardware-friendly functions such as attention has widespread application. Here we aim to make MGLUs efficient on modern hardware accelerators (GPUs) as well. In general, low batch size settings (for simplicity, we consider batch size 1) of LLMs' generative inference require matrix-vector products. Such operations are almost always memory-bound, as the cost of reading the weight matrix dominates the compute time. A naïve PyTorch implementation of MGLUs would issue $n_m$ separate matrix-vector multiplies (and corresponding element-wise Swish and multiplies), resulting in $n_m \times 2 + 1$ full-precision weight memory reads per token—an extremely inefficient pattern.

To address this, we implement a fused CUDA kernel and a simple triton (Tillet et al., 2019) kernel that:

- **Packs mask bits:** Combine the $n_m$ binary masks $M_i$ for each weight entry into a single 8-bit integer, so that one load fetches all mask information for a block of weights.

- **Shared loading:** For each thread block, load a tile of the shared weight matrix $W$ and the corresponding packed mask words in one coalesced transaction for all $n_m \times 2$ output vectors.

This design reduces the number of global memory reads from $n_m \times 2$ to $1$ ($W$ load) $+ 1$ (mask load) per tile, while performing all Swish and element-wise multiplications in fast on-chip memory. As a result, our kernel achieves up to $19.66\times$ speedup over naïve PyTorch implementation on RTX5090 with $n_m = 8$. Algorithm 1 presents the detailed procedure, and Figure 3 illustrates the forward

pass. The core idea is to fetch all necessary weight and packed-mask blocks for each output element in a single coalesced read and complete every arithmetic operation entirely in registers, thereby eliminating redundant global-memory traffic.

## 4.1 Discussion

**Memory Load During Inference.** Table 2 represents the parameter count and required memory load at inference time. Inference in modern transformer FFNs is typically bound by memory bandwidth rather than raw compute. In SwiGLU, the gating and value projections each require reading an FP16 weight matrix of size $h \times d$ for a total of two matrices per token (memory load $16 \times 2hd$ bits), whereas a normal GELU activation only requires one up-projection. SwiMGLU instead folds the two intermediate projections into a single FP16 matrix $W \in \mathbb{R}^{h \times d}$ and applies $n_m$ binary masks $M_i \in \{0,1\}^{h \times d}$ at inference to recover gating. The resulting per-token memory load is one FP16 matrix ($16 \times 2hd$ bits for up-projection) plus $n_m$ mask bits ($n_m\, hd$ bits). For $n_m = 1$, the relative reduction is

$$\frac{16 \cdot 2hd - \left(16 \cdot hd + hd\right)}{16 \cdot 2hd} = 0.46875 \,,$$

*i.e.,* up to 47% fewer bits transferred, directly translating to faster inference on memory-bound hardware. Mixtures with $n_m \leq 16$ therefore still reduces the parameter count of a standard SwiGLU FFN at 16bit inference time, yet retain SwiGLU-level expressivity during optimization.[2]

Table 2: Parameter count and memory-access cost per token for the intermediate layers of FFN variants during FP16 inference. $h$ and $d$ denote the hidden and intermediate sizes, respectively.

| Layer type | #Params (FP16) | #Params (Binary) | Memory Load (bits) |
|---|---|---|---|
| LU | $hd$ | $0$ | $\mathbf{16hd}$ |
| GLU | $2hd$ | $0$ | $\mathbf{32hd}$ |
| MGLU | $hd$ | $n_m hd$ | $(\mathbf{16} + \boldsymbol{n_m})\boldsymbol{hd}$ |

**Number of Parameters.** SwiMGLU replaces two full-rank SwiGLU weight matrices ($2hd$ FP16 values) with one matrix plus $n_m$ masks, yielding a footprint of $16hd + n_m\, hd$ bits at inference (masks stored in 1-bit form). During training, masks are kept as FP16 logits, adding $n_m\, hd$ FP16 parameters.

**Computational Cost.** At inference time (next token prediction), only the forward pass is executed. SwiMGLU incurs $2(1 + n_m)\, hd$ multiply–add operations per token, versus $6hd$ for SwiGLU. On memory-bound hardware the extra FLOPs are negligible: memory bandwidth dominates, and SwiMGLU's reduction in FP16 reads directly lowers inference latency. Training on the other hand is compute bound with the backward costing roughly twice the FLOPs of the forward. Masks are stored as FP16 logits and use straight-through estimation, adding $n_m\, hd$ FP16 parameters and one extra element-wise multiply–add per mask in each pass. Consequently, the total training cost becomes $(6 + 8n_m)\, hd$, compared to $18\, hd$ per token in SwiGLU, introducing a runtime overhead according to the mask size.

## 5 Experiments

We demonstrate the effectiveness and efficiency of SwiMGLU. We selected the Llama 3 architecture (Grattafiori et al., 2024) with SwiGLU as the baseline and compare the different FFN layers.

### 5.1 Setup

**Model Configuration.** Our baseline model follows the Llama family design (Grattafiori et al., 2024). We train models at two scales: a 159M *small* (12 layers, $h = 768$, $d = 3072$) and a 1.08B *large* (16 layers, $h = 2048$, $d = 8192$. For SwiMGLU, we substitute each SwiGLU layer with the SwiMGLU layer, keeping all other architectural components (e.g. attention, normalization, embeddings) identical. Note that SwiMGLU would have a smaller model size as the number of projection weights are reduced.

**Hyperparameters.** All model are trained using the AdamW optimizer ($\beta_1 = 0.9, \beta_2 = 0.99, \epsilon = 1 \times 10^{-8}$) with a learning rate of $3 \times 10^{-4}$ and $1 \times 10^{-4}$ for *small* and *large* models respectively, weight decay of 0.1, linear warmup for the first 10% steps, followed by cosine decay. A full list of hyperparameters and training settings are listed in Appendix A.

---

[2]For Llama-1B ($h = 2048,\ d = 8192$) the FFN weights per layer shrink from 96MB to 64MB, while the binary mask costs only 2MB in boolean form.

Table 3: Zero-shot accuracy (%) on downstream tasks and validation perplexity. #weights represent the number of weight parameters excluding masks bits.

| | $n_m$ | #weights | PPL↓ | ArcE↑ | ArcC↑ | HS↑ | PiQA↑ | SciQ↑ | WG↑ | Avg ↑ |
|---|---|---|---|---|---|---|---|---|---|---|
| GELU | – | 113M | 25.8 | 47.47 | 19.45 | 28.09 | 60.61 | 64.80 | 52.41 | 45.47 |
| SwiGLU | – | 141M | 23.7 | 48.15 | 20.05 | 28.53 | 61.43 | 67.90 | 51.14 | 46.20 |
| SwiMGLU | 1 | 113M | 25.0 | 48.91 | 19.28 | 28.25 | 60.72 | 69.00 | 50.20 | 46.06 |
| SwiMGLU | 2 | 113M | 24.5 | **49.12** | 19.97 | 28.49 | 60.01 | **70.60** | 51.53 | **46.62** |
| SwiMGLU | 4 | 113M | 23.9 | 48.99 | **20.56** | 28.49 | **61.70** | 69.10 | 50.04 | 46.48 |
| SwiMGLU | 8 | 113M | **23.5** | 48.65 | **20.56** | **28.63** | 61.53 | 68.00 | **51.54** | 46.49 |
| SwiGLU | – | 1.08B | **12.3** | 64.94 | **28.92** | 37.20 | 69.15 | **84.50** | 51.30 | 56.00 |
| SwiMGLU | 1 | 808M | 13.0 | 63.72 | 27.73 | 36.20 | 68.61 | 83.00 | 54.30 | 55.59 |
| SwiMGLU | 2 | 808M | 12.7 | 62.08 | 26.71 | 36.52 | 68.44 | 84.20 | 51.85 | 54.97 |
| SwiMGLU | 4 | 808M | 12.4 | **65.78** | **28.92** | **37.69** | **69.26** | 84.20 | **55.25** | **56.85** |

Table 4: Two-shot accuracy (%) on downstream tasks and validation perplexity. #weights represent the number of weight parameters excluding masks bits.

| | $n_m$ | #weights | PPL↓ | ArcE↑ | ArcC↑ | HS↑ | PiQA↑ | SciQ↑ | WG↑ | Avg ↑ |
|---|---|---|---|---|---|---|---|---|---|---|
| GELU | – | 113M | 25.8 | 48.57 | 19.45 | 27.87 | 60.77 | 61.10 | 52.88 | 45.11 |
| SwiGLU | – | 141M | 23.7 | 48.65 | 20.14 | **28.64** | 61.70 | 62.30 | **51.70** | 45.52 |
| SwiMGLU | 1 | 113M | 25.0 | 48.48 | 19.03 | 28.16 | 60.45 | 61.40 | 50.75 | 44.71 |
| SwiMGLU | 2 | 113M | 24.5 | 49.62 | **21.42** | 28.55 | 60.39 | 62.50 | 51.14 | 45.60 |
| SwiMGLU | 4 | 113M | 23.9 | 49.41 | 21.16 | 28.61 | **62.51** | 66.10 | 50.59 | 46.40 |
| SwiMGLU | 8 | 113M | **23.5** | **52.02** | 19.62 | 28.54 | 62.08 | **66.80** | 51.38 | **46.74** |
| SwiGLU | – | 1.08B | **12.3** | 66.16 | 31.48 | 37.25 | 69.21 | 88.30 | 51.78 | 57.36 |
| SwiMGLU | 1 | 808M | 13.0 | 65.57 | 30.20 | 36.25 | **69.59** | 88.50 | 53.83 | 57.32 |
| SwiMGLU | 2 | 808M | 12.7 | 65.07 | **31.66** | 36.32 | 68.77 | **88.70** | 52.17 | 57.11 |
| SwiMGLU | 4 | 808M | 12.4 | **67.13** | 30.55 | **37.53** | 69.48 | 88.60 | **53.91** | **57.87** |

**Datasets and Metrics.** We pre-train both baseline and SwiMGLU models on the FineWeb-Edu 100B dataset (Penedo et al., 2024) with *small* models being trained on a 10B token subset. For downstream evaluation, we report zero-shot and two-shot accuracy on six standard benchmarks: ARC Easy (ArcE) (Clark et al., 2018), ARC Challenge (ArcC) (Clark et al., 2018), HellaSwag (HS) (Zellers et al., 2019), PiQA Bisk et al. (2020), SciQ (Welbl et al., 2017), and Winogrande (WG) (Sakaguchi et al., 2021). We utilize the LM Evaluation Harness (Gao et al., 2024) for standardized performance evaluation.

## 5.2 Downstream Performance

**Downstream Accuracy.** Tables 3 and 4 present the zero-shot and two-shot performance respectively. We observe that SwiMGLU generally matches or outperforms SwiGLU when 4 or more masks are employed. For example $m = 4$ showcases an average of 56.85% compared to SwiGLU's 56.00%. These results highlight the superior performance of model using the SwiMGLU activation function. Increasing the mask count generally improves the model performance.

**Latency.** Figure 4 shows the wall-clock latency of a single projection layer in a single batch setting measured on an RTX 5090 at FP16 precision. We evaluate three implementations: our fused CUDA kernel (Flash-MGLU), a Triton re-implementation, and the naïve PyTorch GLU `nn.Linear` baseline.

For the *large* setting ($h=2048$, $d=8192$) with $n_m = 8$, FlashMGLU cuts the matrix–vector time from **0.5210 ms to 0.0834 ms**, delivering a **19.66×** speed-up over the PyTorch baseline and a $6.25×$ acceleration even on our triton prototype. Importantly, these gains persist at much larger scales: with

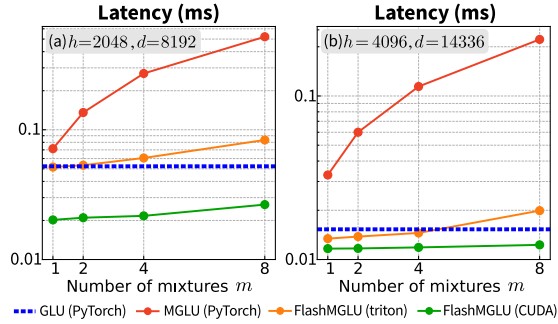

Figure 4: Latency comparison

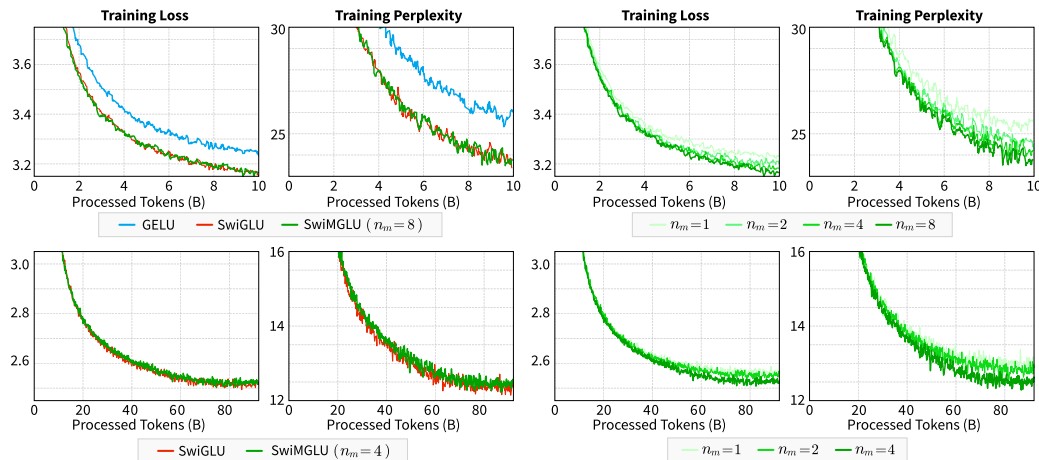

Figure 5: **Comparison of learning curves for different FFN architectures.** The top and bottom rows illustrate the changes in training loss / training perplexity of *small* and *large* models respectively. The left columns compare existing methods against SwiMGLU, and the right columns compare the number of masks.

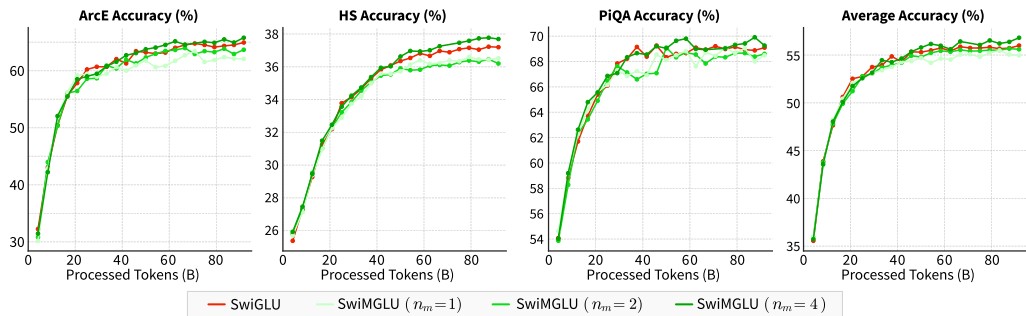

Figure 6: **Comparison of downstream task scores across different FFN architectures of *large* models.** In all metrics, the proposed method, SwiMGLU $n_m = 4$ achieves the best performance.

larger intermediate sizes ($h = 4096$, $d = 14336$) used in larger Llama-3.2 8B models, our optimized CUDA kernel still outpaces the PyTorch baseline by an impressive **18.0×**, and a 11.1× speed-up in our triton prototype.

While the standard GLU implementation must fetch two full-precision weight matrices from global memory, FlashMGLU accesses each weight exactly once and performs all mask operations directly within on-chip registers. By fusing weight loading with mask evaluation, FlashMGLU significantly increases arithmetic intensity, effectively saturating the GPU's Streaming Multiprocessor pipeline and reducing latency by up to **1.51×** under the single-mask variant. As a result, our native CUDA implementation of FlashMGLU consistently outperforms both the naïve MGLU and the standard GLU baselines. Although Triton incurs slightly higher overhead on very small matrices, this gap narrows as the hidden dimensions increase. Moreover, since CUDA allows developers to explicitly control memory access patterns, native CUDA code achieves higher performance overall. A full head-to-head comparison between FlashMGLU and a highly tuned *standard* GLU kernel is presented in Appendix E.

**Scaling with the number of masks.** Under a naïve PyTorch MGLU kernel, runtime scales nearly linearly for $n_m \leq 8$, doubling the number of mask mixtures almost doubles the latency, driven by proportional increases in memory-traffic. In contrast, FlashMGLU's unified weight-and-mask loading strategy decouples execution time from mask count, yielding only marginal slowdowns as $n_m$ increases. Even under a computationally intense eight mask variant, we still observe a **1.15×** speed-up on small models and a **1.24×** acceleration on larger variants. Once $n_m = 8$, register-file pressure begins to spill into local memory, and the performance advantage over the standard GLU kernel gradually erodes, which is an effect that is especially noticeable in our Triton prototype.

## 5.3 Ablations and Analysis

**Training Dynamics.** Figure 5 plots token-level cross-entropy (left axis) and perplexity (right axis) over the first 150k optimization steps for both *small* and *large* models trained with GELU, SwiGLU, and our SwiMGLU variants. Models using SwiMGLU with $n_m = 4$ consistently show comparable or lower losses compared to those using SwiGLU. Figure 6 illustrates the downstream performance on ARC-Easy, HellaSwag, PiQA, and the average accuracy across all downstream tasks. SwiMGLU with $n_m = 4$ outperforms SwiGLU on all tasks, with notable improvements, demonstrating strong generalization capabilities of MoGE despite its reduced parameter count.

**Training Cost.** While SwiMGLU significantly reduces inference-time memory access and latency, it incurs a higher computational cost during pretraining. As shown in Appendix A (Table 6), achieving comparable downstream performance to SwiGLU requires approximately twice the wall-clock training time due to the additional optimization of mask parameters. This trade-off reflects a deliberate design choice: we prioritize lower inference cost and memory footprint at the expense of moderately higher training time. We believe this is a reasonable compromise, as real-world deployments of LLMs are typically dominated by inference workloads rather than pretraining.

**Training Stability.** At default learning rates, both MGLU and SwiGLU converge with comparable smoothness. However, under higher learning rates, where SwiGLU experiences pronounced loss spikes, the masked variants effectively dampen these excursions and recover more rapidly. The four-mask configuration, in particular, proves exceptionally robust. Detailed loss curves for all learning-rate settings are provided in Appendix B.

**Number of Masks $n_m$.** For the *small* model size, increasing the number of masks from $n_m = 1$ to $n_m = 8$ results in a strictly monotonic reduction in training loss. Concurrently, this also improves the two-shot average accuracy, which rises from 44.71% to 46.74%. Nevertheless, the marginal benefit diminishes once $n_m = 8$, while memory and compute overhead continue to increase. Empirically, the $n_m = 4$ setting achieves the optimal balance between quality and resource usage. These findings suggest that a compact collection of complementary masks can fully recover, and occasionally surpass, the expressive power of a dense SwiGLU gating mechanism.

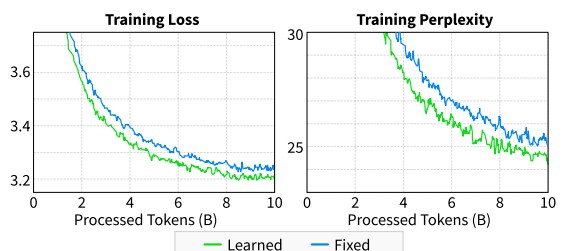

Figure 7: Training loss and perplexity of learned vs. fixed masks in *small* configuration and $n_m = 2$.

**Learned vs. Fixed Masks.** Figure 7 contrasts models in which the masks are co-optimized with the network parameters against models that rely on randomly sampled but *fixed* masks. Allowing the masks to learn consistently tracks a lower training loss throughout and improves the final perplexity by roughly $0.5$. By comparison, freezing the masks essentially collapses the learning curve to that of the single-mask ($n_m = 1$) baseline, underlining that it is *mask-combination learning*—not sparsity alone—that drives the gain in expressivity. Complete results are provided in Appendix B.

**Mask Distribution.** Inspection of the learned binary masks shows that their activation ratio is *not* forcibly balanced: depending on the layer, the proportion of ones can skew either above or below 50% (typically ranging from roughly 45% to 55%). Full statistics for every layer and model size are provided in Appendix C. This flexibility lets the network *learn* how much capacity to devote to the gate versus the value pathway, effectively tuning the gate-value trade-off on a per-layer basis. Because a dense SwiGLU block lacks such adaptive capacity allocation, this mechanism offers a plausible explanation for the cases in which MGLU surpasses SwiGLU in accuracy.

## 6 Conclusion and Future Work

In this work, we introduced Masked Gated Linear Units (MGLUs), a novel family of feed-forward activations that recover the expressivity of traditional GLUs using a single shared weight matrix sculpted by learnable binary masks. Our Mixture of Element-wise Gating (MoEG) design not only matches or exceeds the performance of the commonly used SwiGLU activation on a variety of language-understanding benchmarks, but also delivers substantial efficiency gains at inference time.

In terms of limitations, MGLUs are more computationally expensive than other GLU variants, since they increase arithmetic complexity in exchange for a reduced memory footprint. However, by developing FlashMGLU, an optimized kernel that fuses mask unpacking with the core matrix-vector operations, we achieve up to a $19.66\times$ speedup over a naïve PyTorch MGLU implementation on modern GPUs, while reducing memory-bandwidth requirements and latency compared to standard GLU variants. These advances pave the way for richer gating mechanisms to be deployed in latency- and memory-constrained settings without compromising model quality.

## 7 Acknowledgement

This work was supported by DENSO IT LAB Recognition, Control, and Learning Algorithm Collaborative Research Chair (Science Tokyo).

This work used computational resources TSUBAME4.0 supercomputer provided by Institute of Science Tokyo through the HPCI System Research Project (Project ID: hp240170).

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

# A    Experimental Details

Here we provide more details about the model architecture, training configurations and resources used in our experiments.

## A.1    Model Architecture and Training Configuration

Table 5 summarizes the architectural configurations shared across all experiments. Table 6 lists the number of weight and mask parameters, along with the corresponding estimated storage sizes at inference time for each model variant.

All experiments reported in this paper are implemented based on the `llm-recipes` framework (Fujii et al., 2024). All models are trained on TSUBAME supercomputer with NVIDIA H100 GPUs (94GB), with *small* models trained on 4GPUs and *large* models trained on 16GPUs with Fully Sharded Data Parallel (FSDP). Training GPU hours are shown in Table 6.

Table 5: Model architecture of *small* and *large* variants.

| Model Size | $h$ | $d$ | Context Length | #Heads | #Layers |
|---|---|---|---|---|---|
| *small* | 768 | 3072 | 1024 | 24 | 12 |
| *large* | 2048 | 8192 | 4096 | 32 | 16 |

Table 6: Number of weight and mask parameters, estimated storage size, and training GPU hours for each model configuration.

| Scale | Model | $n_m$ | #Weights | #Masks | Size (MB) | GPU Hours |
|---|---|---|---|---|---|---|
| *small* | GELU | – | 113M | 0 | 215 | 18 |
| | SwiGLU | – | 141M | 0 | 269 | 22 |
| | SwiMGLU | 1 | 113M | 28.3M | 219 | 20 |
| | SwiMGLU | 2 | 113M | 56.6M | 222 | 24 |
| | SwiMGLU | 4 | 113M | 113M | 229 | 33 |
| | SwiMGLU | 8 | 113M | 226M | 242 | 52 |
| *large* | SwiGLU | – | 1.08B | 0 | 2052 | 768 |
| | SwiMGLU | 1 | 808M | 268M | 1573 | 752 |
| | SwiMGLU | 2 | 808M | 537M | 1605 | 1008 |
| | SwiMGLU | 4 | 808M | 1.07B | 1669 | 1376 |

## A.2    Hyperparameters

Table 7 lists the hyperparameters that we use by default at training time for all our experiments.

Table 7: Pretraining hyperparameters for *small* and *large* models.

| | *small* | *large* |
|---|---|---|
| Optimizer | AdamW | AdamW |
| Learning Rate (LR) | 3E-4 | 1E-4 |
| Minimum LR | 3E-5 | 1E-5 |
| LR Schedule | cosine | cosine |
| Weight Decay | 0.1 | 0.1 |
| $\beta_1$ | 0.9 | 0.9 |
| $\beta_2$ | 0.99 | 0.99 |
| $\epsilon$ | 1E-8 | 1E-8 |
| Gradient Clipping | 1 | 1 |
| Global Batch Size | 512 | 512 |
| Warmup Steps | 1000 | 4800 |

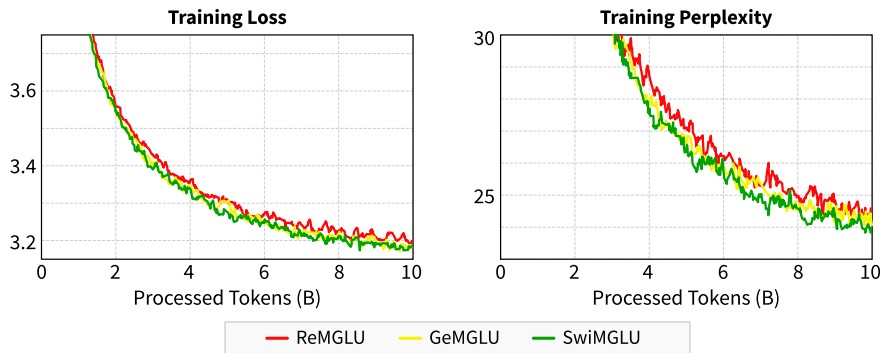

Figure 8: **Trainig Curves of *small* MGLU variants with different activation functions.** Left: training loss; right: validation perplexity.

Table 8: Zero-shot accuracy (%) on downstream tasks and validation perplexity across different activation functions. #weights represent the number of weight parameters excluding masks bits.

| | $n_m$ | #weights | PPL↓ | ArcE↑ | ArcC↑ | HS↑ | PiQA↑ | SciQ↑ | WG↑ | Avg ↑ |
|---|---|---|---|---|---|---|---|---|---|---|
| SwiGLU | – | 141M | **23.7** | 48.15 | 20.05 | **28.53** | 61.43 | 67.90 | 51.14 | 46.20 |
| ReMGLU | 4 | 113M | 24.3 | **49.66** | 19.71 | 28.45 | 61.53 | 69.10 | 49.80 | 46.38 |
| GeMGLU | 4 | 113M | 24.0 | 48.65 | 19.54 | 28.50 | **62.30** | 69.30 | 52.17 | **46.74** |
| SwiMGLU | 4 | 113M | 23.9 | 48.99 | 20.56 | 28.49 | 61.70 | 69.10 | 50.04 | 46.48 |

Table 9: Two-shot accuracy (%) on downstream tasks and validation perplexity across different activation functions. #weights represent the number of weight parameters excluding masks bits.

| | $n_m$ | #weights | PPL↓ | ArcE↑ | ArcC↑ | HS↑ | PiQA↑ | SciQ↑ | WG↑ | Avg ↑ |
|---|---|---|---|---|---|---|---|---|---|---|
| SwiGLU | – | 141M | **23.7** | 48.65 | 20.14 | 28.64 | 61.70 | 62.30 | 51.70 | 45.52 |
| ReMGLU | 4 | 113M | 24.3 | 49.28 | 19.71 | 28.25 | 61.32 | 61.80 | 51.22 | 45.26 |
| GeMGLU | 4 | 113M | 24.0 | 49.28 | 20.56 | **28.74** | 62.13 | 66.40 | 52.25 | **46.56** |
| SwiMGLU | 4 | 113M | 23.9 | **49.41** | 21.16 | 28.61 | **62.51** | 66.10 | 50.59 | 46.40 |

# B    Additional Experiments

## B.1    Other Activation Functions.

In the main paper we concentrated on SwiMGLU, whose gating function relies on the Swish non-linearity used in standard SwiGLU. To verify that the advantages of our mask-based design are not tied to a single activation, this appendix evaluates several alternatives that are widely adopted in large-scale language models—GELU, ReLU, and SiLU. For each activation we replace the gating function $g(\cdot)$ in Eq. (3) while keeping all other architectural choices and training hyper-parameters identical to the baseline. We report both pre-training perplexity and downstream accuracy of *small* models so that we can directly compare the impact of each non-linearity under the same experimental conditions.

**Training Curve.** Figure 8 plots training loss and perplexity for the various activations. ReMGLU converges more slowly than both GeMGLU and SwiMGLU, with SwiMGLU achieving the fastest convergence and slightly outperforming GeMGLU.

**Downstream Evaluation.** Table 8 and Table 9 show the downstream evaluation scores on different activation functions in MGLU. The masked design improves or matches the SwiGLU baseline across all activations while using fewer trained weights, confirming that the capacity unlocked by mask pairs is activation-agnostic. Smoother nonlinearities (GELU and SiLU) benefit most: GeMGLU achieves the best average zero-shot accuracy (46.74%) and ties for the best two-shot average (46.56%), whereas SwiMGLU attains the lowest validation perplexity (23.9). ReMGLU, though slightly weaker than its smoother counterparts, and comparable to the SwiGLU baseline on average accuracy.

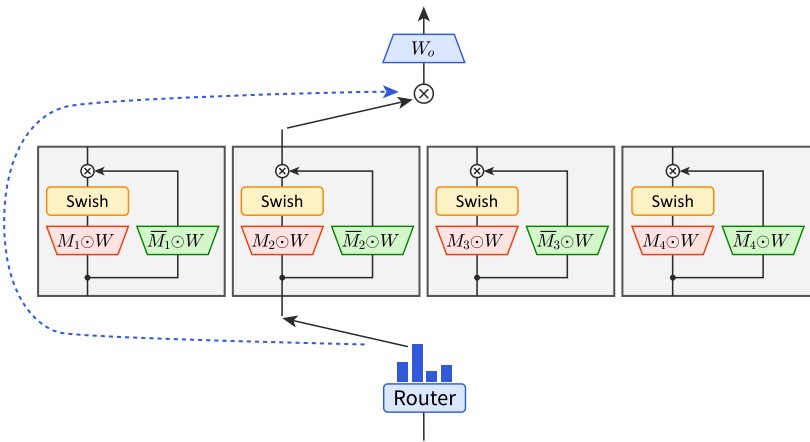

Figure 9: **Diagram of a Top-1 SwiMGLU block.** We illustrate one token being routed across four mask experts, where the router independently routes each token. The Top-1 SwiMGLU layer returns the output of the selected experts multiplied by the router gate value.

## B.2 Top-K Routing

Summation over every masking route maximizes capacity but may be wasteful for tokens that require only a subset of specialized subspaces. Inspired by routing techniques in switch transformers (Fedus et al., 2021), we therefore explore a *Top-K routing* variant of MGLU that, at inference time, activates only the $K$ masks whose logits yield the highest gate magnitudes. The subsection first details the routing algorithm and its lightweight implementation—requiring a single additional linear router over the input features, then evaluates downstream scores as $K$ varies from 1 (fully sparse) to $n_m$ (fully dense). We show that, on *small* configuration, routing with $K = 2$ retains most of the accuracy gains of full MGLU with $n_m = 4$, while reducing computational FLOPs by $2\times$, offering a practical knob for deployments with stringent latency budgets.

**Top-K routed MGLU.** Given an input token representation $\boldsymbol{x} \in \mathbb{R}^h$, a lightweight "router" computes a vector of logits

$$\boldsymbol{\ell} \; = \; \boldsymbol{x}W_r \quad \in \mathbb{R}^{n_m}, \tag{4}$$

where $W_r$ is a learned $h \times n_m$ matrix. Following Fedus et al. (2021), we retain only the $K$ largest logits. Applying a softmax over this truncated vector yields sparse gating weights

$$G(\boldsymbol{x}) \; = \; \mathrm{Softmax}\big(\mathrm{TopK}(\boldsymbol{\ell})\big) \quad \in \mathbb{R}^{n_m}, \tag{5}$$

The Top-K routed MGLU is denoted by

$$\mathrm{MGLU}_{\text{Top-}K}(\boldsymbol{x}) \; = \; \sum_{i=1}^{n_m} G(\boldsymbol{x})_i \, g\big(\boldsymbol{x}(M_i \odot W)\big) \; \odot \; \boldsymbol{x}\big(\overline{M_i} \odot W\big). \tag{6}$$

Because $G(\boldsymbol{x})$ contains at most $K$ non-zero terms, only those $K$ masked projections need be evaluated—reducing memory traffic and latency compared with summing over all $n_m$ routes while preserving most of the accuracy gains of full MGLU. Figure 9 illustrates the overall architecture.

**Training Curves.** Figure 10 presents the training loss and perplexity across different routing configurations. We observe that both Top-4 and Top-2 routed SwiMGLU closely track the loss trajectory of the non-routed SwiMGLU, indicating that sparse routing with multiple active masks retains most of the model's learning capacity. In contrast, Top-1 routing introduces a noticeable performance degradation, suggesting that activating only a single expert per token limits representational power.

**Downstream Evaluation.** Table 10 and Table 11 summarize validation perplexity and task performance for each routing strategy. The **Top-4** router ($K = 4$) consistently dominates, delivering the lowest perplexity in both settings, and the highest average accuracy. The **Top-2** variant ($K = 2$) offers a favorable trade-off: it matches or surpasses the unrouted SwiMGLU ($n_m{=}4$) on most metrics, boosts average zero-shot accuracy to 46.64 %, and retains a competitive two-shot score of 45.79 %.

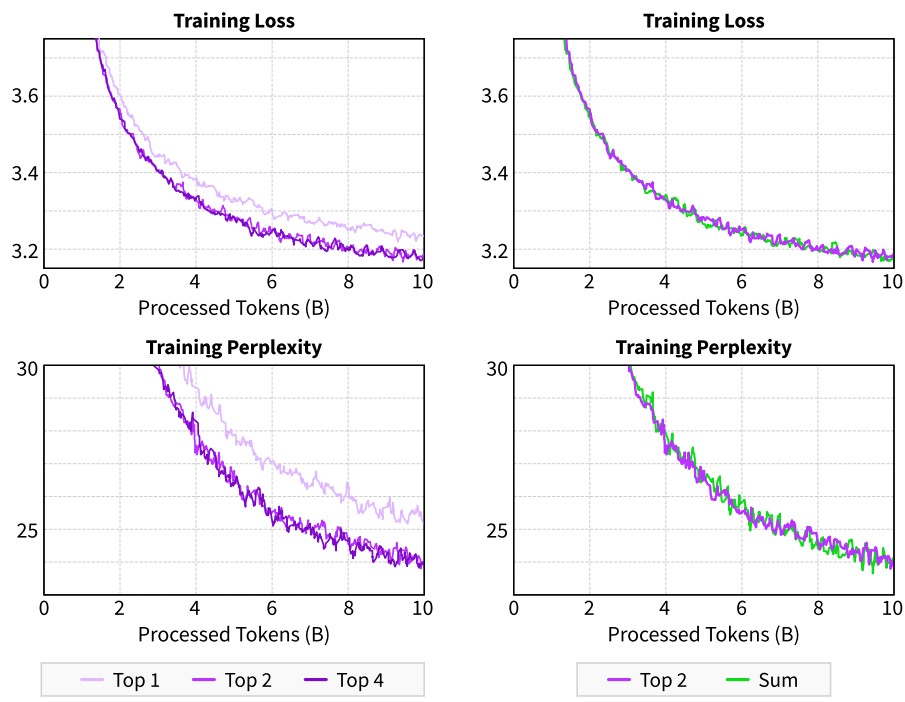

**Figure 10: Learning curves of *small* SwiMGLU models under different Top-$K$ routing strategies.**
Left: training loss and validation perplexity for $K \in \{1, 2, 4\}$; right: Top-2 routing compared with the non-routed SwiMGLU baseline.

Table 10: Zero-shot accuracy (%) on downstream tasks and validation perplexity across different routing coefficient K. #weights represent the number of weight and router parameters excluding masks bits. The boldface and underline indicate, respectively, the best and second-best value per column.

| | $n_m$ | $K$ | #weights | PPL↓ | ArcE↑ | ArcC↑ | HS↑ | PiQA↑ | SciQ↑ | WG↑ | Avg ↑ |
|---|---|---|---|---|---|---|---|---|---|---|---|
| SwiGLU | – | – | 141M | **23.7** | 48.15 | 20.05 | 28.53 | 61.43 | 67.90 | 51.14 | 46.20 |
| SwiMGLU | 4 | – | 113M | 23.9 | 48.99 | 20.56 | 28.49 | **61.70** | 69.10 | 50.04 | 46.48 |
| SwiMGLU | 4 | 1 | 113M | 25.2 | 48.11 | 19.80 | 27.89 | 61.04 | 68.90 | 50.67 | 46.07 |
| SwiMGLU | 4 | 2 | 113M | 24.0 | 48.86 | 20.56 | 28.29 | 60.55 | 69.50 | 52.09 | 46.64 |
| SwiMGLU | 4 | 4 | 113M | 23.8 | **49.20** | **21.84** | **28.70** | 61.37 | **71.80** | **53.43** | **47.72** |

Table 11: Two-shot accuracy (%) on downstream tasks and validation perplexity across different routing coefficient K. #weights represent the number of weight and router parameters excluding mask bits. The boldface and underline indicate, respectively, the best and second-best value per column.

| | $n_m$ | $K$ | #weights | PPL↓ | ArcE↑ | ArcC↑ | HS↑ | PiQA↑ | SciQ↑ | WG↑ | Avg ↑ |
|---|---|---|---|---|---|---|---|---|---|---|---|
| SwiGLU | – | – | 141M | **23.7** | 48.65 | 20.14 | 28.64 | 61.70 | 62.30 | 51.70 | 45.52 |
| SwiMGLU | 4 | – | 113M | 23.9 | 49.41 | 21.16 | 28.61 | **62.51** | **66.10** | 50.59 | 46.40 |
| SwiMGLU | 4 | 1 | 113M | 25.2 | 49.03 | 20.31 | 27.95 | 60.50 | 59.90 | 52.17 | 44.98 |
| SwiMGLU | 4 | 2 | 113M | 24.0 | **51.39** | 20.65 | 28.71 | 62.08 | 60.40 | 51.54 | 45.79 |
| SwiMGLU | 4 | 4 | 113M | 23.8 | 49.79 | **22.27** | **28.82** | 62.13 | 63.00 | **52.57** | **46.43** |

In contrast, **Top-1** routing ($K = 1$) shows a significant drop in both perplexity and average accuracy, suggesting that allocating at least two experts per token is critical for maintaining representation power. Overall, these results indicate that $K = 2$ not only reduces compute but also enhances generalization across a diverse suite of downstream tasks.

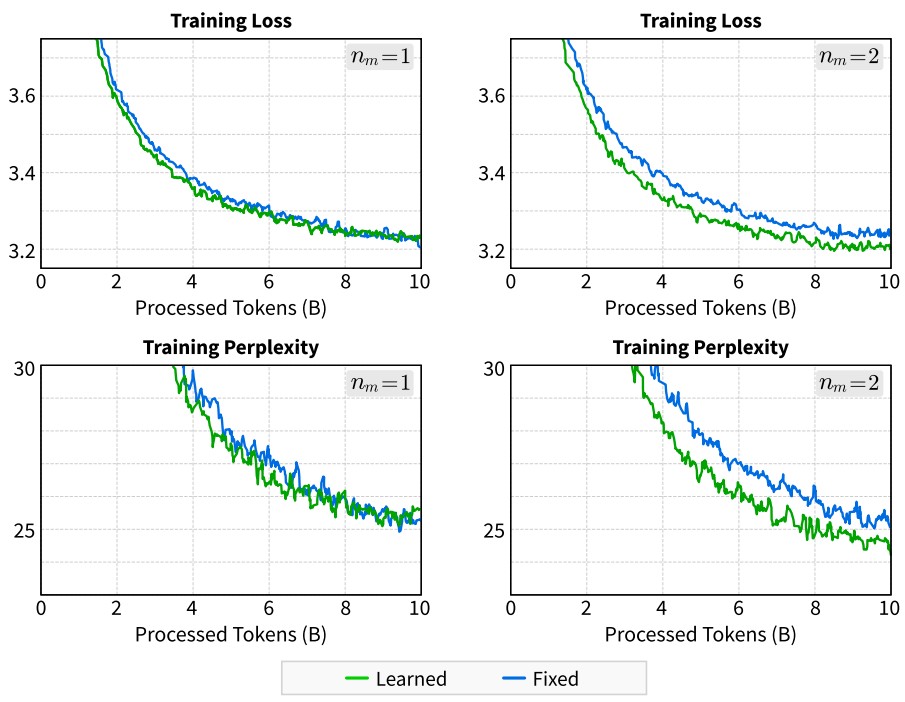

Figure 11: **Training curves of learned vs. fixed masks in *small* SwiMGLU models.** Lef: $n_m = 1$; Right: $n_m = 2$.

Table 12: Zero-shot accuracy (%) on downstream tasks and validation perplexity on *small* models.

|  | $n_m$ | Mask Type | PPL↓ | ArcE↑ | ArcC↑ | HS↑ | PiQA↑ | SciQ↑ | WG↑ | Avg ↑ |
|---|---|---|---|---|---|---|---|---|---|---|
| SwiMGLU | 1 | Learned | 25.0 | 48.91 | 19.28 | 28.25 | 60.72 | 69.00 | 50.20 | 46.06 |
| SwiMGLU | 1 | Fixed | 25.1 | 47.60 | **21.08** | 28.19 | 61.10 | 68.30 | **51.54** | 46.30 |
| SwiMGLU | 2 | Learned | **24.5** | **49.12** | 19.97 | **28.49** | 60.01 | **70.60** | 51.53 | **46.62** |
| SwiMGLU | 2 | Fixed | 25.1 | 48.32 | 19.03 | 28.23 | **61.81** | 67.30 | 50.36 | 45.84 |

Table 13: Two-shot accuracy (%) on downstream tasks and validation perplexity on *small* models.

|  | $n_m$ | Mask Type | PPL↓ | ArcE↑ | ArcC↑ | HS↑ | PiQA↑ | SciQ↑ | WG↑ | Avg ↑ |
|---|---|---|---|---|---|---|---|---|---|---|
| SwiMGLU | 1 | Learned | 25.0 | 48.48 | 19.03 | 28.16 | 60.45 | 61.40 | 50.75 | 44.71 |
| SwiMGLU | 1 | Fixed | 25.1 | 48.11 | 20.14 | 28.02 | 61.21 | 60.30 | 51.78 | 44.92 |
| SwiMGLU | 2 | Learned | **24.5** | **49.62** | **21.42** | **28.55** | 60.39 | **62.50** | 51.14 | **45.60** |
| SwiMGLU | 2 | Fixed | 25.1 | 47.14 | 20.31 | 28.24 | **61.43** | 61.40 | **51.93** | 45.07 |

## B.3 Learned vs. Fixed Masks.

This section compares learned and fixed masks in *small* SwiMGLU models. While previous sections focus on the number and structure of masks, here we investigate the benefit of co-optimizing masks alongside model weights.

**Training Curve.** Figure 11 shows the training loss and perplexity when using learned versus fixed masks. While with $n_m = 2$, learned masks consistently outperform fixed ones, with $n_m = 1$ the difference is subtle.

**Downstream Evaluation.** Table 12 and Table 13 shows the downstream task scores across different mask configurations. The learned mask configuration with $n_m = 2$ also improves downstream accuracy compared to fixed masks, highlighting the importance of mask combination adaptation. These results suggest that expressivity is not solely due to sparsity, but also driven by mask learning with multiple masks.

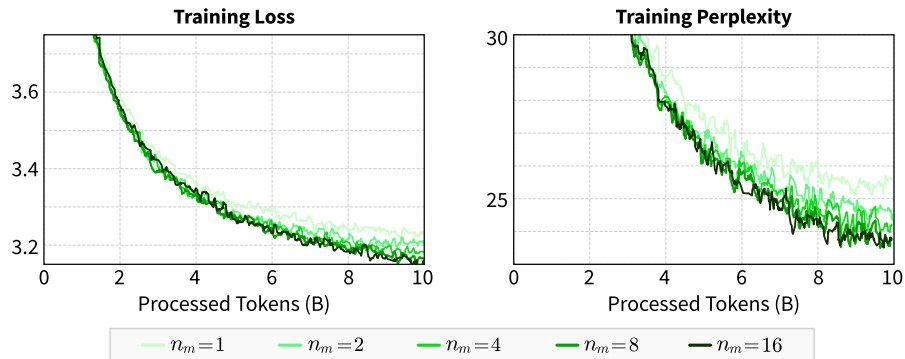

Figure 12: **Training curves of *small* SwiMGLU models across different mask count.**

Table 14: Zero-shot accuracy (%) on downstream tasks and validation perplexity. #weights represent the number of weight parameters excluding masks bits. The boldface and underline indicate, respectively, the best and second-best value per column.

|  | $n_m$ | #weights | PPL↓ | ArcE↑ | ArcC↑ | HS↑ | PiQA↑ | SciQ↑ | WG↑ | Avg ↑ |
|---|---|---|---|---|---|---|---|---|---|---|
| SwiMGLU | 1 | 113M | 25.0 | 48.91 | 19.28 | 28.25 | 60.72 | 69.00 | 50.20 | 46.06 |
| SwiMGLU | 2 | 113M | 24.5 | **49.12** | 19.97 | 28.49 | 60.01 | **70.60** | 51.53 | **46.62** |
| SwiMGLU | 4 | 113M | 23.9 | 48.99 | 20.56 | 28.49 | **61.70** | 69.10 | 50.04 | 46.48 |
| SwiMGLU | 8 | 113M | 23.5 | 48.65 | 20.56 | **28.63** | 61.53 | 68.00 | **51.54** | 46.49 |
| SwiMGLU | 16 | 113M | **23.3** | 48.15 | **21.08** | **28.63** | 61.59 | 68.50 | 50.91 | 46.47 |

Table 15: Two-shot accuracy (%) on downstream tasks and validation perplexity. #weights represent the number of weight parameters excluding masks bits. The boldface and underline indicate, respectively, the best and second-best value per column.

|  | $n_m$ | #weights | PPL↓ | ArcE↑ | ArcC↑ | HS↑ | PiQA↑ | SciQ↑ | WG↑ | Avg ↑ |
|---|---|---|---|---|---|---|---|---|---|---|
| SwiMGLU | 1 | 113M | 25.0 | 48.48 | 19.03 | 28.16 | 60.45 | 61.40 | 50.75 | 44.71 |
| SwiMGLU | 2 | 113M | 24.5 | 49.62 | **21.42** | 28.55 | 60.39 | 62.50 | 51.14 | 45.60 |
| SwiMGLU | 4 | 113M | 23.9 | 49.41 | 21.16 | 28.61 | **62.51** | 66.10 | 50.59 | 46.40 |
| SwiMGLU | 8 | 113M | 23.5 | **52.02** | 19.62 | 28.54 | 62.08 | 66.80 | 51.38 | 46.74 |
| SwiMGLU | 16 | 113M | **23.3** | 50.76 | 21.08 | **28.66** | 61.70 | **66.90** | **52.25** | **46.89** |

## B.4 Scaling $n_m$ to 16

We investigate the impact of increasing the number of masks $n_m$ up to 16. As shown in Figure 12, scaling $n_m$ leads to steady improvements in both training loss and perplexity. However, the benefit diminishes beyond $n_m = 4$, and the gains from $n_m = 8$ to $n_m = 16$ are marginal.

**Downstream Evaluation.** Tables 14 and 15 report zero-shot and two-shot accuracy across different values of $n_m$. The $n_m = 16$ setting achieves the best average accuracy, but the improvements over $n_m = 4$ and $n_m = 8$ are small. These results suggest that increasing the number of complementary masks helps, but overly large mask sets yield diminishing returns.

## B.5 Partial Mask Ablation

In the standard single-mask MGLU layer (Eq. 2), the gate and value streams are computed from complementary subspaces of a shared weight matrix $W$ defined by binary masks $M$ and $\overline{M}$. To assess the necessity of these masks, we introduce three ablation variants that remove the masks from

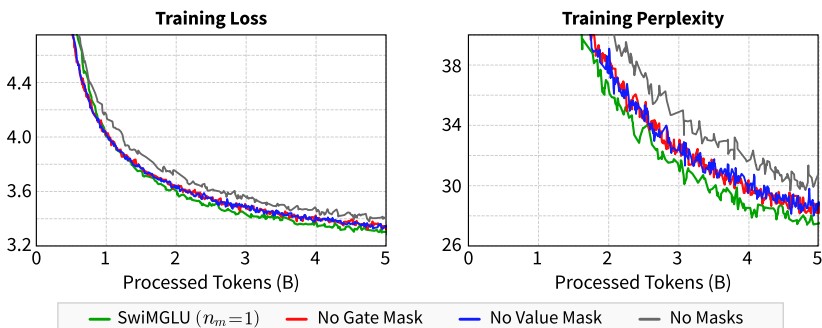

Figure 13: **Training loss and perplexity for mask-ablation variants.** All ablation variants—No Gate Mask, No Value Mask, and No Masks—converge to higher loss and perplexity compared to the fully masked MGLU baseline, highlighting the necessity of maintaining complementary mask-defined subspaces.

gate, value, or both streams:

$$\text{No Gate Mask (Dense Gate)}: \quad h_{\text{NG}}(x) = g\big(xW\big) \odot \big(x(\overline{M} \odot W)\big), \tag{7}$$

$$\text{No Value Mask (Dense Value)}: \quad h_{\text{NV}}(x) = g\big(x(M \odot W)\big) \odot \big(xW\big), \tag{8}$$

$$\text{No Masks (Fully Shared)}: \quad h_{\text{NM}}(x) = g\big(xW\big) \odot \big(xW\big). \tag{9}$$

These variants maintain the multiplicative gate-value interaction characteristic of the MGLU architecture while systematically testing the significance of the mask-defined complementary subspaces.

**Training Curves.** Figure 13 compares training loss and validation perplexity for these mask-ablation variants in the *small* model setting. All ablation variants (No Gate Mask, No Value Mask, and No Masks) show slower convergence and higher final loss compared to the fully masked baseline. These results highlight the importance of distinct, complementary subspaces defined by masks for optimal MGLU performance.

## C   Mask Distribution

To understand how the model allocates capacity when multiple masks are learned, we measure the fraction of rows in the shared projection matrix $W$ that each mask devotes to the *gate* pathway (higher values indicate more gate parameters; the remainder are routed to the value pathway). Figure 14 plots this layer-wise gate ratio for models trained with $n_m = 1, 2$, and $4$.

All configurations exhibit a shallow U-shape: gate capacity is largest in the first layer, reaches a minimum around the middle of the network, and rises again toward the top. Because all masks share the same underlying weight matrix, their ability to partition rows adaptively provides a weight-efficient means of balancing gate and value capacity across depth—one that would be impractical with independently parameterized experts.

## D   PyTorch Implementation of MGLU

PyTorch implementations of the MGLU layer used for training are provided in Algorithm 2.

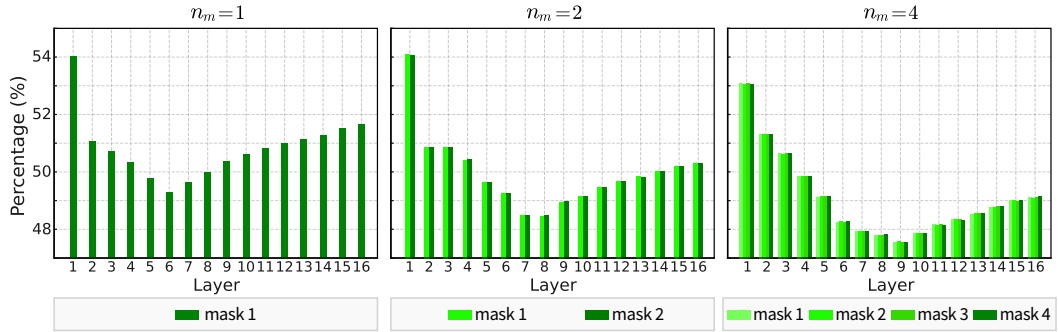

Figure 14: **Layer-wise gate allocation for learned masks.**

---

**Algorithm 2** PyTorch-Style Implementation of MGLU ($n_m = 1$).

---

```python
class MGLU(nn.Linear):
    def __init__(self, in_features, out_features):
        super(MGLU, self).__init__(in_features, out_features, False)
        self.register_parameter(
            "mask", nn.Parameter(0.01 * torch.randn(out_features, in_features), requires_grad=True)
        )

    # convert mask to binary by straight-through estimator
    def ste_mask(self, soft_mask):
        hard_mask = (soft_mask > 0).to(soft_mask.dtype)
        hard_mask = (hard_mask - soft_mask).detach() + soft_mask
        return hard_mask

    def forward(self, x):
        hard_mask = self.ste_mask(self.mask)
        # compute complementary output: e1, e2
        e1 = F.linear(x, self.weight * hard_mask)
        e2 = F.linear(x, self.weight * (1.0 - hard_mask))
        return e1, e2
```

---

# E    Efficient Kernel Implementation

A straightforward PyTorch implementation of a masked-GLU (MGLU) layer must *(i)* load the weight matrix from HBM multiple times—once for every mask bit—and *(ii)* launch several separate `matmul + activation` operations. Because each extra load traverses the bandwidth-limited HBM $\leftrightarrow$ SRAM link, such code quickly becomes *memory-bound*. By fusing the entire computation into a single CUDA kernel we touch each weight exactly *once*, keep partial results in registers, and remove almost all redundant global-memory traffic.

**Implementation Details.** Algorithm 3 shows the memory-access pattern and compute loop for the simplest MGLU case, $n_m = $ `N_MASKS`. Each thread block is launched with coordinates (`row, chunk`) so that it processes one output row and one $K$-slice at a time. Two FP16 weights and activations are fetched together as a single `__half2` load, converted once to `float2`, and kept in registers for the entire multiply–accumulate step. The binary masks are packed eight bits per `int8`; testing the active bit yields a `0/1` scalar that is multiplied into the product. Both the ungated sum and each gated sum accumulate only in registers; after the loop a warp-level shuffle reduces these partials and a single atomic write per row sends the result to HBM.

**Head-room on Hopper.** Although the kernel already eliminates almost all redundant global-memory traffic, it deliberately avoids Hopper-specific optimisations such as `cp.async` and Tensor Memory Accelerator (TMA). Incorporating those features could conservatively deliver an additional $1.2$–$1.3\times$ speed-up on server-grade H100 GPUs—an avenue we leave to future work.

**Latency Comparison.** In Tables 16 and 17, we report the execution latency of our kernel versus a naïve Torch implementation on RTX 5090 and H100 GPUs, respectively; Tables 18 and 19 then compare our kernel against the standard GLU implementation on the same devices.

**Algorithm 3** Simplified CUDA Implementation of MGLU ($n_m$ = N_MASKS).

```
__global__ void mv_splitk_masks_kernel(
    const __half* __restrict__ A, // [M x N], row-major
    const __half* __restrict__ x, // [N]
    const int8_t* __restrict__ mask, // [M x N],
    float* __restrict__ y, // masked outputs
    int M, int N, int split_k
) {
    int row = blockIdx.x; int chunk = blockIdx.y;
    if (row >= M || chunk >= split_k) return;

    // compute [start,end) of this K-chunk
    int chunk_size = (N + split_k - 1) / split_k;
    int start = chunk * chunk_size; int end = min(start + chunk_size, N);
    int row_off = row * N;
    int idx = start + threadIdx.x * 2; int stride = blockDim.x * 2;

    float total = 0.0f; float msum[N_MASKS * 2] = {0.0f, 0.0f, ...};
    for (; idx + 1 < end; idx += stride) {
        __half2 a2 = *reinterpret_cast<const __half2*>(&A[row_off + idx]);
        __half2 x2 = *reinterpret_cast<const __half2*>(&x[idx]);
        float2 af = __half22float2(a2);
        float2 xf = __half22float2(x2);
        float2 prod = { af.x*xf.x, af.y*xf.y };
        total += prod.x + prod.y;

        int8_t m0 = __ldg(&mask[row_off + idx]);
        int8_t m1 = __ldg(&mask[row_off + idx+1]);
        unsigned int b0 = (unsigned int)m0;
        unsigned int b1 = (unsigned int)m1;
        #pragma unroll
        for (int b = 0; b < N_MASKS; ++b) {
            float mb0 = float((b0 >> b) & 1u); float mb1 = float((b1 >> b) & 1u);
            if (b0 & (1u << b)) msum[b] += prod.x; if (b1 & (1u << b)) msum[b] += prod.y;
        }
    }
    // reduce and write to HBM.
```

Table 16: MGLU latency and speed-ups. Torch MGLU is the naïve `nn.Linear` implementation on an RTX 5090 GPU; higher ratios mean faster custom kernels.

| $n_m$ | $h$ | $d$ | CUDA (ms) | Triton (ms) | Torch (ms) | Torch/CUDA | Torch/Triton |
|---|---|---|---|---|---|---|---|
| 1 | 8192 | 2048 | **0.0202** | 0.0516 | 0.0715 | **3.54×** | 1.39× |
| 2 | 8192 | 2048 | **0.0210** | 0.0533 | 0.1358 | **6.47×** | 2.55× |
| 4 | 8192 | 2048 | **0.0217** | 0.0606 | 0.2715 | **12.51×** | 4.48× |
| 8 | 8192 | 2048 | **0.0265** | 0.0834 | 0.5210 | **19.66×** | 6.25× |
| 1 | 14336 | 4096 | **0.1166** | 0.1342 | 0.3289 | **2.82×** | 2.45× |
| 2 | 14336 | 4096 | **0.1169** | 0.1381 | 0.6001 | **5.13×** | 4.35× |
| 4 | 14336 | 4096 | **0.1186** | 0.1454 | 1.1426 | **9.63×** | 7.86× |
| 8 | 14336 | 4096 | **0.1229** | 0.1990 | 2.2172 | **18.04×** | 11.14× |

Table 17: MGLU latency and speed-ups. Torch MGLU is the naïve `nn.Linear` implementation on a H100 GPU; higher ratios mean faster custom kernels.

| $n_m$ | $h$ | $d$ | CUDA (ms) | Triton (ms) | Torch (ms) | Torch/CUDA | Torch/Triton |
|---|---|---|---|---|---|---|---|
| 1 | 8192 | 2048 | **0.0395** | 0.0639 | 0.1296 | **3.28×** | 2.03× |
| 2 | 8192 | 2048 | **0.0409** | 0.0679 | 0.2213 | **5.41×** | 3.26× |
| 4 | 8192 | 2048 | **0.0428** | 0.0810 | 0.4117 | **9.62×** | 5.08× |
| 8 | 8192 | 2048 | **0.0483** | 0.1294 | 0.7910 | **16.38×** | 6.11× |
| 1 | 14336 | 4096 | **0.1044** | 0.1191 | 0.3896 | **3.73×** | 3.27× |
| 2 | 14336 | 4096 | **0.1067** | 0.1239 | 0.7080 | **6.64×** | 5.71× |
| 4 | 14336 | 4096 | **0.1110** | 0.1608 | 1.3454 | **12.12×** | 8.37× |
| 8 | 14336 | 4096 | **0.1215** | 0.3202 | 2.6354 | **21.68×** | 8.23× |

Table 18: CUDA MGLU latency and speed-ups against the standard PyTorch GLU baseline (no masking) on an RTX 5090 GPU. Lower latency and higher speed-up are better.

| $n_m$ | $h$ | $d$ | CUDA MGLU (ms) | Torch GLU | GLU/CUDA |
|---|---|---|---|---|---|
| 1 | 8192 | 2048 | **0.0202** | 0.0306 | **1.51×** |
| 2 | 8192 | 2048 | **0.0210** | 0.0306 | **1.46×** |
| 4 | 8192 | 2048 | **0.0217** | 0.0306 | **1.41×** |
| 8 | 8192 | 2048 | **0.0265** | 0.0306 | **1.15×** |
| 1 | 14336 | 4096 | **0.1166** | 0.1530 | **1.31×** |
| 2 | 14336 | 4096 | **0.1169** | 0.1530 | **1.31×** |
| 4 | 14336 | 4096 | **0.1186** | 0.1530 | **1.29×** |
| 8 | 14336 | 4096 | **0.1229** | 0.1530 | **1.24×** |

Table 19: CUDA MGLU latency and speed-ups against the standard PyTorch GLU baseline (no masking) on a H100 GPU. Lower latency and higher speed-up are better.

| $n_m$ | $h$ | $d$ | CUDA MGLU (ms) | Torch GLU | GLU/CUDA |
|---|---|---|---|---|---|
| 1 | 8192 | 2048 | **0.0395** | 0.0485 | **1.23×** |
| 2 | 8192 | 2048 | **0.0409** | 0.0485 | **1.19×** |
| 4 | 8192 | 2048 | **0.0428** | 0.0485 | **1.13×** |
| 8 | 8192 | 2048 | **0.0483** | 0.0485 | **1.00×** |
| 1 | 14336 | 4096 | **0.1044** | 0.1215 | **1.16×** |
| 2 | 14336 | 4096 | **0.1067** | 0.1215 | **1.14×** |
| 4 | 14336 | 4096 | **0.1110** | 0.1215 | **1.09×** |
| 8 | 14336 | 4096 | **0.1215** | **0.1215** | **1.00×** |

# F  Broader Impacts

The method proposed in this paper will not lead to negative societal impact. By reducing memory-bandwidth pressure and per-token latency, it cuts the energy consumption of large-language-model inference and therefore lowers carbon emissions. These efficiency gains also make advanced language capabilities viable on commodity and edge hardware, broadening access while trimming operational costs.

