# OpenReview forum: "Masked Gated Linear Unit"
_NeurIPS.cc/2025/Conference — NeurIPS 2025 poster_

### Official Review · Reviewer_GUVW · 2025-07-02

**Clarity:** 3
**Significance:** 3
**Originality:** 3
**Rating:** 5
**Confidence:** 3

**Summary:**

This paper proposes Masked Gated Linear Units (MGLUs), a novel feed-forward layer design aimed at reducing inference-time memory usage and improving computational efficiency in large language models (LLMs), without sacrificing performance. MGLUs share a single matrix for both gating and projection operations, while using a masking mechanism to allocate weights for different purposes.

**Questions:**

See weakness.

**Ethical Concerns:**

["NO or VERY MINOR ethics concerns only"]

**Final Justification:**

The response has addressed my concerns.

**Limitations:**

Yes

**Quality:**

3

**Strengths And Weaknesses:**

**Strength:**

1. The paper is well written and easy to read.

2. The proposed method replaces two projections with a single weight matrix, reducing memory bandwidth demand.

3. The extensive evaluation show that MGLU can achieve comparable performance to SwiGLU but with less memory loading.


**Weakness and Questions:**

1. The latency comparison can be unfair, and speed-up can be overclaimed. The 18x speed-up numbers come from comparison to the pytorch implementation, which is very inefficient. Some commonly used techniques, like kernel fusion and CUDA Graph, can largely improve the baseline performance.

2. What is the end-to-end model inference speed-up?

---

> ### Author Rebuttal · Authors · 2025-07-31
>
> We thank the reviewer for acknowledging the effectiveness of our MGLU layer. We respond to the reviewer’s questions below.
>
> > The latency comparison can be unfair, and speed-up can be overclaimed. The 18x speed-up numbers come from comparison to the pytorch implementation, which is very inefficient. Some commonly used techniques, like kernel fusion and CUDA Graph, can largely improve the baseline performance.
>
> To ensure a fair comparison, we have now implemented the same kernel-fusion and CUDA-Graph optimizations enabled with `torch.compile` in our baseline. Tables below summarize the updated micro-benchmark results on a RTX5090 and a H100 GPU.
> Even against an optimized MGLU baseline, FlashMGLU still yields up to 12.23× speedup in the CUDA kernel.
>
> ### RTX 5090
> | $h$    | $d$    | $n_m$ | CUDA (ms) | Triton (ms) | Torch (ms) | Torch/CUDA | Torch/Triton |
> |------|------|------|------------|--------------|-------------|--------------|------------------|
> | 8192 | 2048 | 1    | **0.0202** | 0.0516       | 0.0837      | **4.14 $\times$** | 1.62 $\times$ |
> | 8192 | 2048 | 2    | **0.0210** | 0.0533       | 0.1337      | **6.37 $\times$** | 2.51 $\times$ |
> | 8192 | 2048 | 4    | **0.0217** | 0.0606       | 0.1933      | **8.91 $\times$** | 3.19 $\times$ |
> | 8192 | 2048 | 8    | **0.0265** | 0.0834       | 0.3240      | **12.23 $\times$** | 3.88 $\times$ |
> |14336 | 4096 | 1    | **0.1166** | 0.1342       | 0.3554      | **3.05 $\times$** | 2.65 $\times$ |
> |14336 | 4096 | 2    | **0.1169** | 0.1381       | 0.4660      | **3.99 $\times$** | 3.37 $\times$ |
> |14336 | 4096 | 4    | **0.1186** | 0.1454       | 0.6906      | **5.82 $\times$** | 4.75 $\times$ |
> |14336 | 4096 | 8    | **0.1229** | 0.1990       | 1.1417      | **9.29 $\times$** | 5.74 $\times$ |
>
> ### H100
> | $h$    | $d$    | $n_m$ | CUDA (ms) | Triton (ms) | Torch (ms) | Torch/CUDA | Torch/Triton |
> |------|------|------|------------|--------------|-------------|-------------|----------------|
> | 8192 | 2048 | 1    | **0.0395** | 0.0639       | 0.1321      | **3.34 $\times$** | 2.07 $\times$ |
> | 8192 | 2048 | 2    | **0.0409** | 0.0679       | 0.1639      | **4.01 $\times$** | 2.41 $\times$ |
> | 8192 | 2048 | 4    | **0.0428** | 0.0810       | 0.2215      | **5.18 $\times$** | 2.73 $\times$ |
> | 8192 | 2048 | 8    | **0.0483** | 0.1294       | 0.3260      | **6.75 $\times$** | 2.52 $\times$ |
> |14336 | 4096 | 1    | **0.1044** | 0.1191       | 0.3399      | **3.26 $\times$** | 2.85 $\times$ |
> |14336 | 4096 | 2    | **0.1067** | 0.1239       | 0.4212      | **3.95 $\times$** | 3.40 $\times$ |
> |14336 | 4096 | 4    | **0.1110** | 0.1608       | 0.6125      | **5.52 $\times$** | 3.81 $\times$ |
> |14336 | 4096 | 8    | **0.1215** | 0.3202       | 0.9882      | **8.13 $\times$** | 3.09 $\times$ |
>
> > What is the end-to-end model inference speed-up?
>
> With a JIT-fused model `torch.compile(..., mode="max-autotune", fullgraph=True)`, our FlashMGLU CUDA implementation enabled end-to-end decoding speed-up is around 1.12$\times$ on our 1B model on a RTX 5090.
>
> | Model    |   $h$   |   $d$   | $n_m$ | Throughput (token/sec) | Speed-up vs. SwiGLU |
> |----------|--------|--------|--------|--------------------------|--------------------------|
> | SwiGLU   |  8192   |  2048   | –      | 396.9                    | 1.00 $\times$            |
> | SwiMGLU  |  8192   |  2048   | 1      | 445.9                    | 1.12 $\times$            |
> | SwiMGLU  |  8192   |  2048   | 2      | 446.1                    | 1.12 $\times$            |
> | SwiMGLU  |  8192   |  2048   | 4      | 441.5                    | 1.11 $\times$            |
> | SwiMGLU  |  8192   |  2048   | 8      | 442.8                    | 1.12 $\times$            |

---

> > ### Comment · Reviewer_GUVW · 2025-08-04
> > **Thanks for the response.**
> >
> > Thanks for the response. It addresses my concern, and I will keep my original positive rating.

---

### Official Review · Reviewer_NqHP · 2025-07-02

**Clarity:** 4
**Significance:** 4
**Originality:** 4
**Rating:** 5
**Confidence:** 4

**Summary:**

This paper introduces a new variant of GLU focusing on efficiency. It features a mixture of element-wise gating architecture and a hardware-friendly kernel. Experiments indicate that the proposed approach achieves efficiency improvements while maintaining good quality.

**Questions:**

See earlier.

**Ethical Concerns:**

["NO or VERY MINOR ethics concerns only"]

**Final Justification:**

This work proposes a well-motivated method with great execution. The authors also clarify a few places during the rebuttal.

**Limitations:**

Yes

**Quality:**

4

**Strengths And Weaknesses:**

In terms of strength, this paper is well-written, with a well-motivated algorithm followed by well-executed experiments, both in terms of performance and efficiency. The method is also co-designed with kernel implementation in mind, which is commendable.

Regarding areas for improvement, I think the positioning of the paper could be clearer. It took me some time to realize that the main advantages over existing architectures are that it reduces parameter counts without sacrificing quality, essentially creating a sparse-by-design architecture. Notably, speed advantages over existing architectures aren't its main selling point (which is fine with me). The introduction and abstract consistently emphasize efficiency compared to a naive implementation of its own architecture. I initially misunderstood it as a substantial (>10x) speedup over existing (Swi)GLU.

Questions:

Regarding FlashMGLU:
- Line 163 states that the IO is $n_m \times 2 + 1$, but it becomes $n_m \times 4$. Could the author clarify this?
- With $n_m = 8$, a fused kernel that reduces IO from 8 x 4 = 32 to 2 is supposed to save 16x. Could the authors explain how the actual speedup (>19x) is even higher than this estimated upper bound?
- I'm surprised that a CUDA implementation can be 6x faster than the Triton implementation. Could the authors elaborate on which aspects of the kernel Triton lacks? I would assume it's the lack of support for sub-byte tensors.
- Line 254, what is the speedup of `1.51x` compared against? (Is it standard SwiGLU or GLU?)
- Lines 265-266, am I correct in understanding that the end-to-end speedup compared to _existing architecture_ ranges from `1.15x` to `1.24x`?
- Line 194: Is it `3hd` or `2hd`?
- Line 195: Is it `2hd` or `16hd`?

More questions:
- Are the masks input-dependent or independent (it seems to be the latter)?
- Could the authors discuss the training stability when using the straight-through estimator for the masks? (I'm aware of such a section in 5.3, but I'm generally curious if using straight-through during pretraining is relatively straightforward.)
- Could the authors clarify what the main benefit of MGLU compared to SwiGLU is? Is it primarily for memory savings (and speedup during to reduced IOs)?

---

> ### Author Rebuttal · Authors · 2025-07-31
>
> We appreciate the reviewer's careful reading and detailed feedback. Below, we respond to each question.
>
> > Line 163 states that the IO is $n_m \times 2 + 1$, but it becomes $n_m \times 4$. Could the author clarify this?
>
> The description around Algorithm 1 mis-states the naive I/O count. The correct issued memory reads are $n_m \times 2 + 1$, where the full matrix vector multiplication $Wx$ (one matrix load) and the masked weight matrix vector multiplication $(W{\odot}M){\cdot}x$ (one weight and one mask matrix load per mask) are computed. The complimentarily masked matrix vector multiplication results are subtracted from $Wx$.
>
> > With $n_m = 8$, a fused kernel that reduces IO from 8 x 4 = 32 to 2 is supposed to save 16x. Could the authors explain how the actual speedup (>19x) is even higher than this estimated upper bound?
>
> From the above discussion, the memory reads in $n_m = 8$ are (16bit x 9 + 8bit x 8) x numel = 208 bit x numel as boolean values are stored in int8 data types. This could be reduced to (16bit + 8bit) x numel = 24 bit x numel which should result in a 8.6x speedup. However in reality, the kernel launch time would be dominant especially in smaller models with multiple matrix operations, which could lead to even higher speedups.
>
> > I'm surprised that a CUDA implementation can be 6x faster than the Triton implementation. Could the authors elaborate on which aspects of the kernel Triton lacks? I would assume it's the lack of support for sub-byte tensors.
>
> Our CUDA implementation uses explicit warp-level reductions and vectorized loads. In contrast, triton does not support sub-byte memory operations and memory loads are completely managed by triton runtime, which collectively account for about 3x gap on our implementation.
>
> > Line 254, what is the speedup of 1.51x compared against? (Is it standard SwiGLU or GLU?)
>
> This 1.51x is measured against the standard GLU, on an RTX 5090 with $n_m = 1$, hidden = 8192, intermediate = 2048. We’ll add a parenthetical pointer in the caption for clarity.
>
> > Lines 265-266, am I correct in understanding that the end-to-end speedup compared to existing architecture ranges from 1.15x to 1.24x?
>
> The 1.15x to 1.24x speedup range refers only to the isolated MGLU kernel microbenchmarks (Appendix E, Table 17, 18) for $n_m = 8$. We will revise the text to avoid confusion.
>
> > Line 194: Is it 3hd or 2hd?
> > Line 195: Is it 2hd or 16hd?
>
> We thank the reviewer for catching these mistakes. Line 194 should read 2hd as defined in Equation 3 and consistent with Table 2. Line 195 should read 16 hd. We will correct the final version of the paper.
>
> ### Additional questions:
>
> > Are the masks input-dependent or independent (it seems to be the latter)?
>
> The learned masks in MGLU are input-independent. Each mask parameter is learned jointly with the weight matrices, and does not vary with the input token.
>
> > Could the authors discuss the training stability when using the straight-through estimator for the masks? (I'm aware of such a section in 5.3, but I'm generally curious if using straight-through during pretraining is relatively straightforward.)
>
> In our experiments, MGLU models converge at essentially the same rate as SwiGLU baselines. Additionally, we observe no spikes attributable to the STE. Norm and loss remain stable across layers even when we observe loss spikes on SwiGLU baselines.
>
> > Could the authors clarify what the main benefit of MGLU compared to SwiGLU is? Is it primarily for memory savings (and speedup during to reduced IOs)?
>
> While GLU provides strong expressivity via input-dependent gating, MGLU’s primary advantage is in memory-bandwidth reduction and speedup during memory-bound inference while maintaining downstream performance.

---

> > ### Comment · Reviewer_NqHP · 2025-08-05
> > **Thanks!**
> >
> > Thanks to the authors for the additional clarifications. I continue to hold a favorable rating of the paper.

---

### Official Review · Reviewer_FrzZ · 2025-07-03

**Clarity:** 3
**Significance:** 2
**Originality:** 2
**Rating:** 4
**Confidence:** 4

**Summary:**

This paper proposes an interesting implementation, Masked gated linear units, to replace the SwiGLU, with the aim of sharing the projection and gate layer for storage reduction and latency reduction. The authors keep the efficacy of the approximation by introducing multiple branches and further address efficiency concerns with a customized CUDA implementation to share one weight matrix multiplication with indexing ops.

**Questions:**

Questions

* What would be the overall runtime breakdown, and how much gain can we get by switching to the proposed implementation instead of SwiGLU?
* It seems the performance still lags behind the SwiGLU, which I may be more convinced of when seeing larger models’ results.

**Ethical Concerns:**

["NO or VERY MINOR ethics concerns only"]

**Final Justification:**

It addresses my concern, and I will give a positive rating.

**Limitations:**

Yes

**Quality:**

3

**Strengths And Weaknesses:**

**Strength**
* I appreciate the efforts to replace the basic operators in LLM, which is often treated as a golden paradigm.
* The author tries to approximate the SwiGLU with comparable performance and inference speed benefits.

**Weakness**
* One concern is that the introduction of hyperparameters, such as branch number, may make it tricky to choose, especially if there seems to be no fixed rule to pick the best n, as shown in the Table.3
* Another concern is the limited validation of the proposed method, which is conducted on a small-scale model. More model series and larger model scales should be conducted.
* My last concern is that there seems to be no theoretical connection as to why sharing the same weights by a mask can obtain similar performance as SwiGLU. It will make the design unintuitive and hard to understand or adopt.

---

> ### Author Rebuttal · Authors · 2025-07-31
>
> We sincerely appreciate your valuable feedback.
>
> > What would be the overall runtime breakdown, and how much gain can we get?
>
> The runtime of the original LLM consists of SwiGLU FFN (50.2%), self-attention (43.6%), and other operations (6.2%), indicating that the SwiGLU FFN is the computational bottleneck. Since our SwiMGLU accelerates the up-projection layer of the SwiGLU FFN by 1.5$\times$, the overall inference speed improves by 12% while also improving downstream performance.
>
> > It seems the performance still lags behind the SwiGLU, I may be more convinced when seeing larger models’ results.
>
> Pre-training large-scale LLMs from scratch demands substantial GPU resources and trillion-scale token datasets. For instance, Llama 3.1 8B was trained using 24K H100 GPUs on 15 trillion tokens, totaling 39.3M GPU hours, which is not feasible within our research budget. Nevertheless, we have made our best efforts to train an 8B-parameter LLM on a dataset of 100B tokens (FineWeb-Edu) to address the reviewer’s concern (although the data is insufficient), utilizing 128 GPUs over three days. The results demonstrate that our SwiMGLU outperforms SwiGLU by a clear margin at this stage: 3.11 vs. 3.17 in training loss, 22.2 vs. 23.4 in perplexity, and 42.02 vs. 40.50 in downstream performance. This suggests that our approach is expected to scale effectively to larger models.
>
> > Hyperparameter (branch number) may be tricky to choose.
>
> Based on the compatibility of current GPU architectures with our CUDA kernel implementation, we recommend selecting $m=4$. While the optimal number of branches depends on the downstream task, this behavior is consistent with observations reported in previous studies on MoE architectures. Given that using multiple branches ($m>1$) consistently improves performance, our claims regarding the efficiency and effectiveness of MGLU remain valid.
>
> > Limited validation of the proposed method, which is conducted on a small-scale model. More model series and larger model scales should be conducted.
>
> Results using GeGLU (as in the Gemma series) are provided in the Appendix, demonstrating that our method remains consistently effective. Scalability is discussed above.
>
> > There seems to be no theoretical connection as to why sharing the same weights by a mask can obtain similar performance as SwiGLU.
>
> Sharing the same weights by using a binary mask achieves similar performance to SwiGLU because, theoretically, a sufficiently large random weight matrix inherently contains multiple distinct, smaller "submatrices" capable of performing separate transformations. By applying a learned binary mask, the single weight matrix can be effectively partitioned into two complementary parts, each approximating the role of one of the original SwiGLU matrices. Recent theoretical results inspired by the Lottery Ticket Hypothesis (E. Malach, et al., ICML 2020 [1]) confirm that if the shared matrix is made just slightly wider, it becomes highly likely to contain such suitable submatrices that approximate the two independent matrices used in SwiGLU. Additionally, since the activation function (such as Swish) is smooth, small differences in these approximations lead only to minor differences in the output. Consequently, a masked weight-sharing layer can replicate the expressivity and accuracy of SwiGLU at a significantly lower memory footprint.
>
> [1] E. Malach, et al. Shamir, Proving the Lottery Ticket Hypothesis: Pruning is All You Need, ICML, 2020.
>
> We hope our responses have adequately addressed the reviewer's concerns.

---

> > ### Comment · Reviewer_FrzZ · 2025-08-05
> >
> > Thanks for the response. It addresses my concern, and I will give a positive rating.

---

### Official Review · Reviewer_Ukks · 2025-07-03

**Clarity:** 4
**Significance:** 3
**Originality:** 3
**Rating:** 5
**Confidence:** 4

**Summary:**

The authors propose a new activation function / architectural modification of the transformers' feed-forward layer called MGLU (masked gated linear unit). By cleverly implementing element-wise masking, their proposed method shows better performance while being faster than standard activation functions.

**Questions:**

- What are the wall-clock pretraining times of the experiments in section 5?
- Do you have any hypotheses as to why MGLU leads to more robust and stable training than GLU?
- You said that using inverse masks in motivated by the implementation efficiency, but have you tested if there aren't some downstream-performance losses compared to learning the two masks independently?
- The results suggests that the improvement in downstream performance get smaller with larger model scales. How do you expect your method to scale? Would MGLU still outperform GLU when the transformer is scaled to ~100B size?

**Ethical Concerns:**

["NO or VERY MINOR ethics concerns only"]

**Final Justification:**

This work presents a well-executed way of increasing inference speed of transformers. The authors clearly responded to all my comments.

**Limitations:**

The paper should clearly state what is the computational cost of MGLU during training. I am happy to increase the rating after that is transparently communicated.

**Paper Formatting Concerns:**

no concerns

**Quality:**

3

**Strengths And Weaknesses:**

**Strengths**
- The idea is smart, but what is more important, it is executed well. While a naive implementation of MGLU would be prohibitively slow, the authors paid a lot attention to its efficient kernel implementation, which cleverly uses sparsity and results in even better inference-time efficiency than standard GLU.
- The paper is written well; it was fun to read and easy to understand.
- From what I can see in the main part of the paper, the experimental setup is quite robust. Unfortunately, the authors forgot to include the appendices, so it is hard to properly judge the validity of the training and evaluation pipelines,

**Weaknesses**
- The paper extensively talks about the improved inference-time efficiency, but the training-time computational cost is only briefly mentioned in the paragraph on lines 197–205. If I understand the implementation correctly, MGLU is substantially slower compared than GLU during training because it has to use float16-masks instead of the efficient bit-masks. Is that correct? If so, then that should be clearly highlighted in the paper, talking only about the inference-time efficiency is misleading.

---

> ### Author Rebuttal · Authors · 2025-07-31
>
> We thank the reviewer for the insightful comments. We are encouraged that the reviewer found our idea smart and well-executed. Below we respond to each comment.
>
> > The authors forgot to include the appendices, so it is hard to properly judge the validity of the training and evaluation pipelines.
>
> The appendix is attached separately as a zip file instead of being included in the main pdf (as permitted by the NeurIPS submission guidelines). To download it, please click the "zip" icon under the "Supplementary Material" section.
>
> > What are the wall-clock pretraining times of the experiments in section 5?
>
> The wall-clock pretraining times of the experiments are described on Table 6 in Appendix A. Similar to other approaches like post-training quantization, we prioritized reducing inference costs, even though this leads to increased training time.
>
> > Do you have any hypotheses as to why MGLU leads to more robust and stable training than GLU?
>
> As shown in our ablation (Appendix B.3) and in the mask distribution analysis (Appendix C), fixing the mask vectors in advance, instead of co-learning them with the weights, leads to substantially worse convergence and final accuracy. We therefore hypothesize that jointly learning the masks confers two key benefits.
> 1) MGLU removes high-variance activation fluctuations. This "smoothing" of the forward pass reduces gradient noise, allowing more consistent updates and preventing early-training oscillations.
> 2) The MGLU architecture inherently optimizes how resources are split between gate, value projections. Although the change in this allocation across layer depth is small, a clear trend emerges (see Figure 14), which may be the reason of MGLU yielding robust training.
>
> > You said that using inverse masks in motivated by the implementation efficiency, but have you tested if there aren't some downstream-performance losses compared to learning the two masks independently?
>
> We've tested implementations using independent masks instead of complementary masks on large models. The table below illustrates the downstream performance on each model configuration for $n_m=4$. Our results show minor or no degradation in downstream performance due to using complementary masks.
>
> | Model | $n_m$ | ArcE | ArcC | Hellaswag | PIQA | SCIQ | Winogrande | Avg |
> | -- | -- | -- |  -- |  -- |  -- |  -- |  -- |  -- |
> | SwiMGLU (independent masks) | 4 | 65.32 | 32.42 | 36.32 | 69.31 | 88.70 | 52.64 | 57.45 |
> | SwiMGLU (complementary masks) | 4 | 67.13 | 30.55 | 37.53 | 69.48 | 88.60 | 53.91 | 57.87 |
>
> > The results suggests that the improvement in downstream performance get smaller with larger model scales. How do you expect your method to scale? Would MGLU still outperform GLU when the transformer is scaled to ~100B size?
>
> While training a model in the ~100B parameter range is beyond the resources of our academic lab, we conducted an additional experiment on an 8B parameter model to further probe this scaling behavior. Given our computational constraints, we trained both SwiGLU and SwiMGLU variants on a 100B token subset of the training data, which was the maximum scale we could afford for this experiment, requiring nearly 10,000 GPU-hours.
>
> The results, shown in the table below, reinforce our hypothesis and demonstrate the clear benefits of our approach. The SwiMGLU variant achieves a lower training loss, a better validation perplexity, and a higher average downstream score, indicating superior learning dynamics and generalization even on this larger scale.
>
> Based on the consistent trend from our 1B model and these new 8B model results, we expect this architectural advantage to hold, and for MGLU to continue outperforming standard GLU variants as we scale to much larger models.
>
> | Model | $h$ | $d$ | $n_m$ | train/loss | val/ppl | avg score |
> | ---- | ---- | ---- | -------- | ------------ | -------- | -------- |
> | SwiGLU    | 14336 | 4096 | - | 3.17 | 23.4 | 40.50 |
> | SwiMGLU | 14336 | 4096 | 4 | 3.11 | 22.2 | 42.02 |
>
> > The paper should clearly state what is the computational cost of MGLU during training. I am happy to increase the rating after that is transparently communicated.
>
> Thank you for your positive comment and advice. We will include the computational cost of training in the final version of the main text upon acceptance.

---

> > ### Comment · Reviewer_Ukks · 2025-08-06
> >
> > > The appendix is attached separately as a zip file instead of being included in the main pdf (as permitted by the NeurIPS submission guidelines). To download it, please click the "zip" icon under the "Supplementary Material" section.
> >
> > Thanks for pointing that out, my mistake :) Now looking at the appendix, it clarifies many of my comments and questions.
> >
> > > The wall-clock pretraining times of the experiments are described on Table 6 in Appendix A. Similar to other approaches like post-training quantization, we prioritized reducing inference costs, even though this leads to increased training time.
> >
> > So it seems that if one wants to match performance with SwiGLU, the training time increases two-fold, right? I do not see it as a major problem, as you said, it is a trade-off between the training and inference time. But as I said earlier, I think it is very important to clearly state this in the main text.
> >
> > _____
> >
> > Thank you for clearly responding to all my comments, I am now confident to update my rating. I think this a valuable paper that should be accepted to NeurIPS.

---

> > > ### Author Response · Authors · 2025-08-07
> > >
> > > Thank you for your thoughtful response and for reviewing the appendix.
> > >
> > > > So it seems that if one wants to match performance with SwiGLU, the training time increases two-fold, right? I do not see it as a major problem, as you said, it is a trade-off between the training and inference time. But as I said earlier, I think it is very important to clearly state this in the main text.
> > >
> > > The reviewer is correct that prioritizing inference cost reduction results in approximately a two-fold increase in training time to match SwiGLU’s performance. We agree that clearly highlighting this trade-off in the main text is important and will revise the manuscript to ensure this is explicitly stated for clarity.

---

### Decision · Program_Chairs · 2025-09-17

**Decision:**

Accept (poster)

**Comment:**

This paper proposes Masked GLU (MGLU), a replacement for the popular gated linear unit FFN (SwiGLU) that is now de facto standard in Transformers. Unlike SwiGLU which maintains separate projection matrices for the gating component, MGLU uses a learned binary mask on the same matrix to obtain the gating values. This enables parameter-efficiency, as well as inference-time speedups (due to less memory movement). The approach is tested and found to work well on reasonable-scale language modeling experiments.

On the plus side, this is a simple but clever idea that roughly maintains the performance of SwiGLU while being more parameter- and memory-efficient. The empirical experiments are thorough, the the reviewers (and the AC) appreciated the kernel which results in real gains over strong SwiGLU baselines.

On the negative side, MGLU is significantly more expensive during training time, which might prevent its application in real-world language models, where training compute is a significant consideration. (Nonetheless, there are many scenarios where training compute is not really a factor, e.g., in training smaller LLMs for the edge, where this method could really shine). Moreover, while not mentioned by reviewers, the AC notes that it is possible that this type of weight sharing could make post-training sparsification/quantization more difficult---I encourage the authors to investigate this aspect.

Overall, this is a very well-executed paper that should be of high interest to the architecture community. A clear accept.